# AD-PT: Autonomous Driving Pre-Training with Large-scale Point Cloud Dataset

**Jiakang Yuan**[1,*], **Bo Zhang**[2,†], **Xiangchao Yan**[2], **Tao Chen**[1,†], **Botian Shi**[2], **Yikang Li**[2], **Yu Qiao**[2]

[1]School of Information Science and Technology, Fudan University
[2]Shanghai Artificial Intelligence Laboratory
jkyuan22@m.fudan.edu.cn, zhangbo@pjlab.org.cn, eetchen@fudan.edu.cn

## Abstract

It is a long-term vision for Autonomous Driving (AD) community that the perception models can learn from a large-scale point cloud dataset, to obtain unified representations that can achieve promising results on different tasks or benchmarks. Previous works mainly focus on the self-supervised pre-training pipeline, meaning that they perform the pre-training and fine-tuning on the same benchmark, which is difficult to attain the performance scalability and cross-dataset application for the pre-training checkpoint. In this paper, for the first time, we are committed to building a large-scale pre-training point-cloud dataset with diverse data distribution, and meanwhile learning generalizable representations from such a diverse pre-training dataset. We formulate the point-cloud pre-training task as a semi-supervised problem, which leverages the few-shot labeled and massive unlabeled point-cloud data to generate the unified backbone representations that can be directly applied to many baseline models and benchmarks, decoupling the AD-related pre-training process and downstream fine-tuning task. During the period of backbone pre-training, by enhancing the scene- and instance-level distribution diversity and exploiting the backbone's ability to learn from unknown instances, we achieve significant performance gains on a series of downstream perception benchmarks including Waymo, nuScenes, and KITTI, under different baseline models like PV-RCNN++, SECOND, CenterPoint. Project page: https://jiakangyuan.github.io/AD-PT.github.io/.

## 1 Introduction

LiDAR sensor plays a crucial role in the Autonomous Driving (AD) system due to its high quality for modeling the depth and geometric information of the surroundings. As a result, a lot of studies aim to achieve the AD scene perception via LiDAR-based 3D object detection baseline models, such as CenterPoint [32], PV-RCNN [17], and PV-RCNN++ [19].

Although the 3D object detection models can help autonomous driving recognize the surrounding environment, the existing baselines are hard to generalize to a new domain (such as different sensor settings or unseen cities). A long-term vision of the autonomous driving community is to develop a scene-generalizable pre-trained model, which can be widely applied to different downstream tasks. To achieve this goal, researchers begin to leverage the Self-Supervised Pre-Training (SS-PT) paradigm recently. For example, ProposalContrast [31] proposes a contrastive learning-based approach to enhance region-level feature extraction capability. Voxel-MAE [6] designs a masking and reconstructing task to obtain a stronger backbone network.

---

*This work was done when Jiakang Yuan was an intern at Shanghai Artificial Intelligence Laboratory.
†Corresponding to Tao Chen and Bo Zhang.

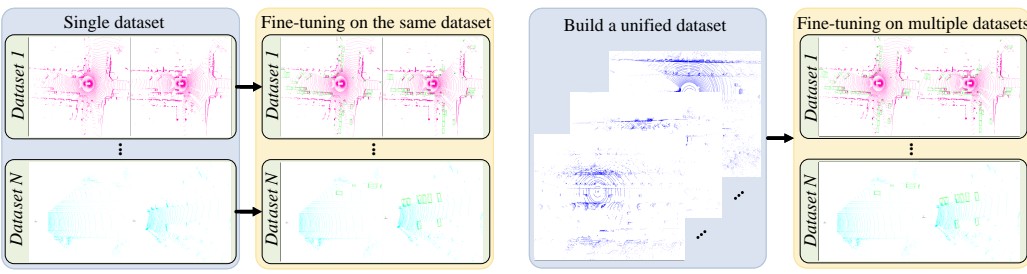

(a) Previous pre-training paradigm.    (b) AD-PT pre-training paradigm.

Figure 1: Differences between previous pre-training paradigm and the proposed AD-PT paradigm.

However, as illustrated in Fig.1, it should be pointed out that there is a crucial difference between the above-mentioned SS-PT and the desired Autonomous Driving Pre-Training (AD-PT) paradigm. The SS-PT aims to learn from a single set of unlabeled data to generate suitable representations for the **same** dataset, while AD-PT is expected to learn unified representations from as large and diversified data as possible, so that the learned features can be easily transferred to various downstream tasks. As a result, SS-PT may only perform well when the test and pre-training data are sampled from the same single dataset (such as Waymo [20] or nuScenes [1]), while AD-PT presents better generalized performance on different datasets, which can be continuously improved with the increase of the number of the pre-training dataset.

Therefore, this paper is focused on achieving the AD-related pre-training which can be easily applied to different baseline models and benchmarks. By conducting extensive experiments, we argue that there are two key issues that need to be solved for achieving the real AD-PT: 1) how to build a unified AD dataset with diverse data distribution, and 2) how to learn generalizable representations from such a diverse dataset by designing an effective pre-training method.

For the first item, we use a large-scale point cloud dataset named ONCE [14], consisting of few-shot labeled (*e.g.*, $\sim$0.5%) and massive unlabeled data. First, to get accurate pseudo labels of the massive unlabeled data that can facilitate the subsequent pre-training task, we design a class-wise pseudo labeling strategy that uses multiple models to annotate different semantic classes, and then adopt semi-supervised methods (*e.g.*, MeanTeacher [21]) to further improve the accuracy on the ONCE validation set. Second, to get a unified dataset with diverse raw data distribution from both LiDAR beam and object sizes, inspired by previous works [29, 24, 33, 34], we exploit point-to-beam playback re-sampling and object re-scaling strategies to diversify both scene- and region-level distribution.

For the second item, we find that the taxonomy differences between the pre-training ONCE dataset and different downstream datasets are quite large, resulting in that many hard samples with taxonomic inconsistency are difficult to be accurately detected during the fine-tuning stage. As a result, taxonomy differences between different benchmarks should be considered when performing the backbone pre-training. Besides, our study also indicates that during the pre-training process, the backbone model tends to fit with the semantic distribution of the ONCE dataset, impairing the perception ability on downstream datasets having different semantics. To address this issue, we propose an unknown-aware instance learning to ensure that some background regions on the pre-training dataset, which may be important for downstream datasets, can be appropriately activated by the designed pre-training task. Besides, to further mine representative instances during the pre-training, we design a consistency loss to constrain the pre-training representations from different augmented views to be consistent.

Our contributions can be summarized as follows:

1. For the first time, we propose the AD-PT paradigm, which aims to learn unified representations by pre-training a general backbone and transfers knowledge to various benchmarks.
2. To enable the AD-PT paradigm, we propose a diversity-based pre-training data preparation procedure and unknown-aware instance learning, which can be employed in the backbone pre-training process to strengthen the representative capability of extracted features.
3. Our study provides a more unified approach, meaning that once the pre-trained checkpoint is generated, it can be directly loaded into multiple perception baselines and benchmarks. Results further verify that such an AD-PT paradigm achieves large accuracy gains on different benchmarks (*e.g.*, 3.41%, 8.45%, 4.25% on Waymo, nuScenes, and KITTI).

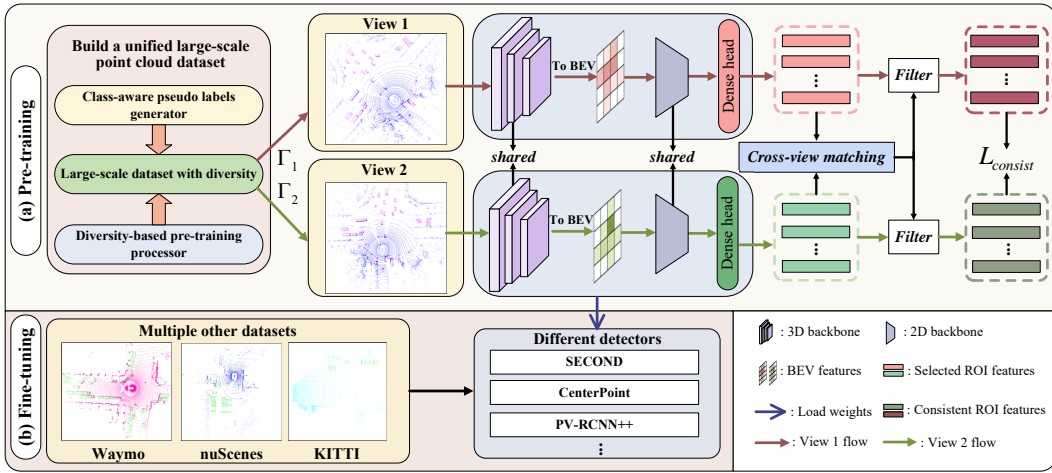

Figure 2: The overview of the proposed AD-PT. By leveraging the proposed method to train on the unified large-scale point cloud dataset, we can obtain well-generalized pre-training parameters that can be applied to multiple datasets and support different baseline detectors.

## 2 Related Works

### 2.1 LiDAR-based 3D Object Detection

Current prevailing LiDAR-based 3D object detection works [11, 16, 17, 19, 27, 18, 4] can be roughly divided into point-based methods, voxel-based methods, and point-voxel-based methods. Point-based methods [16, 30] extract features and generate proposals directly from raw point clouds. PointRCNN [16] is a prior work that generates 3D RoIs with foreground segmentation and designs an RCNN-style two-stage refinement. Unlike point-based methods, voxel-based methods [27, 32] first transform unordered points into regular grids and then extract 3D features using 3D convolution. As a pioneer, SECOND [27] utilizes sparse convolution as 3D backbone and greatly improves the detection efficiency. CenterPoint [32] takes care of both accuracy and efficiency and proposes a one-stage method. To take advantage of both point-based and voxel-based methods, PV-RCNN [17] and PV-RCNN++ [19] propose a point-voxel set abstraction to fuse point and voxel features.

### 2.2 Autonomous Driving-related Self-Supervised Pre-Training

Inspired by the success of pre-training in 2D images, self-supervised learning methods have been extended to LiDAR-based AD scenarios [6, 13, 12, 28, 26, 10, 31]. Previous methods mainly focus on using contrastive learning or masked autoencoder (MAE) to enhance feature extraction. Contrastive learning-based methods [12, 31, 7] use point clouds from different views or temporally-correlated frames as input, and further construct positive and negative samples. STRL [7] proposes a spatial-temporal representation learning that employs contrastive learning with two temporally-correlated frames. GCC-3D [12] and ProposalContrast constructs a consistency map to find the correspondence between different views, developing contrastive learning strategies at region-level. CO3 [2] utilizes LiDAR point clouds from the vehicle- and infrastructure-side to build different views. MAE-based [6, 10] methods utilize different mask strategies and try to reconstruct masked points by a designed decoder. Voxel-MAE [6] introduces a voxel-level mask strategy and verifies the effectiveness of MAE. BEV-MAE [13] designs a BEV-guided masking strategy. More recently, GD-MAE [28] and MV-JAR [26] explore masking strategies in transformer architecture. Different from the existing works that use a designed self-supervised approach to pre-train on unlabeled data and then fine-tune on labeled data with the same dataset, AD-PT aims to pre-train on a large-scale point cloud dataset and fine-tune on multiple different datasets (*i.e.*, Waymo [20], nuScenes[1], KITTI [5]).

Table 1: Performance using different detectors on ONCE validation set. We report mAP using ONCE evaluation metric.

| Detector | Head Choice | Vehicle | Pedestrian | Cyclist |
|---|---|---|---|---|
| ONCE Benchmark (Best) | Center Head | 66.79 | 49.90 | 63.45 |
| CenterPoint (ours) | Center Head | - | **56.01** | - |
| PV-RCNN++ (ours) | Anchor Head | **82.50** | - | **71.19** |

Table 2: Statistics on the number of pseudo-labeled instances per frame. We compare it with ONCE labeled data set.

| ONCE labeled set | | | Pseudo label set | | |
|---|---|---|---|---|---|
| Vehicle | Ped. | Cyclist | Vehicle | Ped. | Cyclist |
| 19.01 | 4.52 | 5.63 | 15.67 | 1.63 | 1.90 |

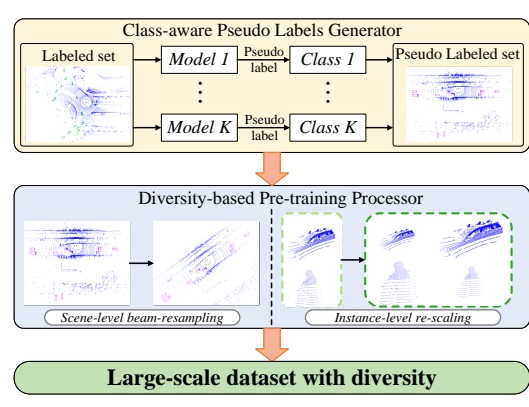

Figure 3: Overall dataset preparation procedure.

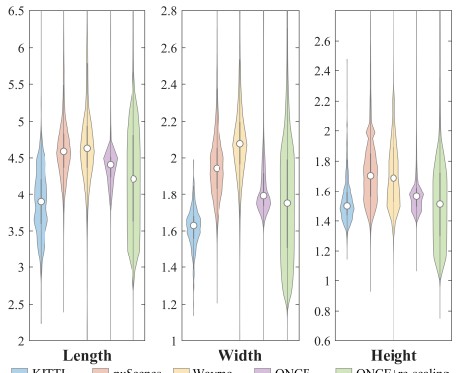

Figure 4: Statistics of object re-scaling.

# 3 Method

To better illustrate our AD-PT framework, we first briefly describe the problem definition and overview of the proposed method in Sec. 3.1. Then, we detail the data preparation and model design of AD-PT in Sec. 3.2 and Sec. 3.3, respectively.

## 3.1 Preliminary

**Problem Formulation.** Different from previous self-supervised pre-training methods, AD-PT performs large-scale point cloud pre-training in a **semi-supervised manner**. Specifically, we have access to $N$ samples which are composed of a labeled set $D_L = \{(x_i, y_i)\}_{i=1}^{N_L}$ and a unlabeled set $D_U = \{x_i\}_{i=1}^{N_U}$, where $N_L$ and $N_U$ denote the number of labeled and unlabeled samples. Note that $N_L$ can be much smaller than $N_U$ (*e.g.*, ∼5K *v.s.* ∼1M, about 0.5% labeled samples). The purpose of our work is to pre-train on a large-scale point cloud dataset with few-shot $D_L$ and massive $D_U$, such that the pre-trained backbone parameters can be used for many down-stream benchmark datasets or 3D detection models.

**Overview of AD-PT.** As shown in Fig. 2, AD-PT mainly consists of a large-scale point cloud dataset preparation procedure and a unified AD-focused representation learning procedure. To initiate the pre-training, a **class-aware pseudo labels generator** is first developed to generate the pseudo labels of $D_U$. Then, to get more diverse samples, we propose a **diversity-based pre-training processor**. Finally, in order to pre-train on these pseudo-labeled data to learn their generalizable representations for AD purposes, an **unknown-aware instance learning** coupled with a consistency loss is designed.

## 3.2 Large-scale Point Cloud Dataset Preparation

In this section, we detail the preparation of the unified large-scale dataset for AD. As shown in Fig. 3, our data creation consists of a class-aware pseudo label generator and a diversity-based pre-training processor to be introduced below.

### 3.2.1 Class-aware Pseudo Labels Generator

It can be observed from Tab. 7 that, pseudo-labels with high accuracy on the pre-training dataset are beneficial to enhance the detection accuracy on downstream datasets such as Waymo [20] and nuScenes [1]. Therefore, to get more accurate pseudo labels, we design the following procedure.

**Class-aware Pseudo Labeling.** ONCE benchmark[3] is utilized to evaluate the pseudo labeling accuracy and more results are shown in the supplementary material. We find that different baseline

---

[3]https://once-for-auto-driving.github.io/benchmark.html

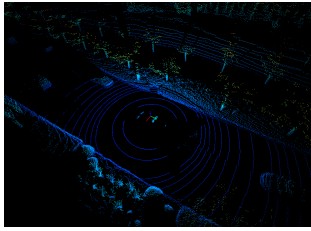 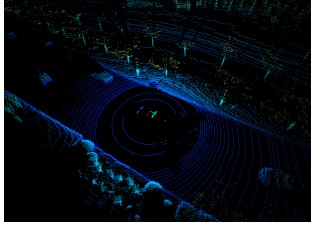 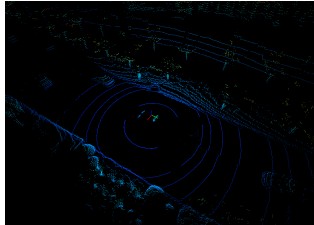

(a) Original point clouds      (b) Upsampled point clouds      (c) Downsampled point clouds

Figure 5: Visualization of point-to-beam playback re-sampling.

models have different biased perception abilities for different classes. For example, the center-based method (*i.e.*, CenterPoint [32]) tends to detect better for small-scale targets (*e.g.*, Pedestrian), while anchor-based methods perform better for other classes. According to the observation, we utilize the PV-RCNN++ [19] with anchor-head to annotate Vehicle and Cyclist classes on ONCE, where the accuracy on Vehicle and Cyclist is found to be much better than the methods listed in the ONCE benchmark. On the other hand, CenterPoint [32] is employed to label the Pedestrian class on ONCE. Finally, we use PV-RCNN++ and CenterPoint to perform a class-wise pseudo labeling process.

**Semi-supervised Data Labeling.** We further employ the semi-supervised learning method to fully exploit the unlabeled data to boost the accuracy of the pseudo labels. As shown in ONCE benchmark, MeanTeacher [21] can better improve the performance on ONCE, and therefore, we use the MeanTeacher to further enhance the class-wise detection ability. Tab. 1 shows that the accuracy on ONCE validation set can be improved after leveraging massive unlabeled data, significantly surpassing all previous records.

**Pseudo Labeling Threshold.** To avoid labeling a large number of false positive instances, we set a relatively high threshold. In detail, for Vehicle, Pedestrian, and Cyclist, we filter out bounding boxes with confidence scores below 0.8, 0.7, and 0.7. As a result, it can be seen from Tab. 2 that, compared with ONCE labeled data, some hard samples with relatively low prediction scores are not annotated.

### 3.2.2 Diversity-based Pre-training Processor

As mentioned in [15, 9], the diversity of data is crucial for pre-training, since highly diverse data can greatly improve the generalization ability of the model. The same observation also holds in 3D pre-training. However, the existing datasets are mostly collected by the same LiDAR sensor within limited geographical regions, which impairs the data diversity. Inspired by [33, 34], discrepancies between different datasets can be categorized into scene-level (*e.g.*, LiDAR beam) and instance-level (*e.g.*, object size). Thus, we try to increase the diversity from the LiDAR beam and object size, and propose a point-to-beam playback re-sampling and an object re-scaling strategy.

**Data with More Beam-Diversity: Point-to-Beam Playback Re-sampling.** To get beam-diverse data, we use the range image as an intermediate variable for point data up-sampling and down-sampling. Specifically, given a LiDAR point cloud with $n$ beam (*e.g.*, 40 beam for ONCE dataset) and $m$ points per ring, the range image $R^{n \times m}$ can be obtained by the following equation:

$$r = \sqrt{x^2 + y^2 + z^2}, \ \ \theta = arctan(x/y), \ \ \phi = arcsin(z/r), \tag{1}$$

where $\phi$ and $\theta$ are the inclination and azimuth of point clouds, respectively, and $r$ denotes the range of the point cloud. Each column and row of the range image corresponds to the same azimuth and inclination of the point clouds, respectively. Then, we can interpolate or sample over rows of a range image, which can also be seen as LiDAR beam re-sampling. Finally, we reconvert the range image to point clouds as follows:

$$x = rcos(\phi)cos(\theta), \ \ y = rcos(\phi)sin(\theta), \ \ z = rsin(\phi), \tag{2}$$

where $x, y, z$ denote the Cartesian coordinates of points, and Fig. 5 shows that point-to-beam playback re-sampling can generate scenes with different point densities, improving the scene-level diversity.

**Data with More RoI-Diversity: Object Re-scaling.** According to the statement in [34, 29, 23] that different 3D datasets were collected in different locations, the object size has inconsistent distributions. As shown in Fig. 4, a single dataset such as ONCE cannot cover a variety of object-size distributions,

resulting in that the model cannot learn a unified representation. To overcome such a problem, we propose an object re-scaling mechanism that can randomly re-scale the length, width and height of each object. In detail, given a bounding box and points within it, we first transform the points to the local coordinate and then multiply the point's coordinates and the bounding box size by a provided scaling factor. Finally, we transform the scaled points with the bounding box to the ego-car coordinate. It can be seen from Fig. 4 that after object re-scaling, the produced dataset contains object sizes with more diversified distributions, further strengthening the instance-level point diversity.

## 3.3 Learning Unified Representations under Large-scale Point Cloud Dataset

By obtaining a unified pre-training dataset using the above-mentioned method, the scene-level and instance-level diversity can be improved. However, unlike 2D or vision-language pre-training datasets which cover a lot of categories to be identified for downstream tasks, our pseudo dataset has limited category labels (*i.e.*, Vehicle, Pedestrian and Cyclist). Besides, as mentioned in Sec. 3.2.1, in order to get accurate pseudo annotations, we set a high confidence threshold, which may inevitably ignore some hard instances. As a result, these ignored instances, which are not concerned in the pre-training dataset but may be seen as categories of interest in downstream datasets (*e.g.*, Barrier in the nuScenes dataset), will be suppressed during the pre-training process.

To mitigate such a problem, it is necessary that both pre-training-related instances and some unknown instances with low scores can be considered when performing the backbone pre-training. From a new perspective, we regard the pre-training as an open-set learning problem. Different from traditional open-set detection [8] which aims at detecting unknown instances, our goal is to activate as many foreground regions as possible during the pre-training stage. Thus, we propose a two-branch unknown-aware instance learning head to avoid regarding potential foreground instances as the background parts. Further, a consistency loss is utilized to ensure the consistency of the calculated corresponding foreground regions.

**Overall Model Structure.** In this part, we briefly introduce our pre-training model structure. Following the prevailing 3D detectors [17, 3, 27], as shown in Fig. 2, the designed pre-training model consists of a voxel feature extractor, a 3D backbone with sparse convolution, a 2D backbone and the proposed head. Specifically, given point clouds $\mathbf{P} \in \mathcal{R}^{N \times (3+d)}$, we first transform points into *different views* through different data augmentation methods $\Gamma_1$ and $\Gamma_2$. Then, voxel features are extracted by a 3D backbone and mapped to BEV space. After that, dense features generated by a 2D backbone can be obtained, and finally, the dense features are fed into the proposed head.

**Unknown-aware Instance Learning Head.** Inspired by previous open-set detection works [8, 35], we consider *background region proposals* with relatively high objectness scores to be unknown instances that are ignored during the pre-training stage but may be crucial to downstream tasks, where the objectness scores are obtained from the Region Proposal Network (RPN). However, due to that these unknown instances contain a lot of background regions, directly treating these instances as the foreground instances during the pre-training will cause the backbone network to activate a large number of background regions. To overcome such a problem, we utilize a two-branch head as a committee to discover which regions can be effectively presented as foreground instances. Specifically, given the RoI features $\mathbf{F}^{\Gamma_1} = [f_1^{\Gamma_1}; f_2^{\Gamma_1}; ...; f_N^{\Gamma_1}] \in \mathcal{R}^{N \times C}$, $\mathbf{F}^{\Gamma_2} = [f_1^{\Gamma_2}; f_2^{\Gamma_2}; ...; f_N^{\Gamma_2}] \in \mathcal{R}^{N \times C}$ and their corresponding bounding boxes $\mathbf{B}^{\Gamma_1} \in \mathcal{R}^{N \times 7}$, $\mathbf{B}^{\Gamma_2} \in \mathcal{R}^{N \times 7}$, where $N$ is the number of RoI features and $C$ denotes the dimension of features, we first select $M$ features $\tilde{\mathbf{F}}^{\Gamma_1} \in \mathcal{R}^{M \times C}$, $\tilde{\mathbf{F}}^{\Gamma_2} \in \mathcal{R}^{M \times C}$ and its corresponding bounding boxes $\tilde{\mathbf{B}}^{\Gamma_1} \in \mathcal{R}^{N \times 7}$, $\tilde{\mathbf{B}}^{\Gamma_2} \in \mathcal{R}^{N \times 7}$ with the highest scores. Then, to obtain the positional relationship corresponding to the activation regions of the two branches, we calculate the distance of the box center between $\tilde{\mathbf{B}}^{\Gamma_1}$ and $\tilde{\mathbf{B}}^{\Gamma_2}$, and the feature correspondence can be obtained by the following equation:

$$(\hat{\mathbf{F}}^{\Gamma_1}, \hat{\mathbf{F}}^{\Gamma_2}) = \{(\tilde{f}_i^{\Gamma_1}, \tilde{f}_j^{\Gamma_2}) | \sqrt{(c_{i,x}^{\Gamma_1} - c_{j,x}^{\Gamma_2})^2 + (c_{i,y}^{\Gamma_1} - c_{j,y}^{\Gamma_2})^2 + (c_{i,z}^{\Gamma_1} - c_{j,z}^{\Gamma_2})^2} < \tau\}, \qquad (3)$$

where $(c_{i,x}^{\Gamma_1}, c_{i,y}^{\Gamma_1}, c_{i,z}^{\Gamma_1})$ and $(c_{j,x}^{\Gamma_2}, c_{j,y}^{\Gamma_2}, c_{j,z}^{\Gamma_2})$ denote $i$-th and $j$-th box center of $\tilde{\mathbf{B}}^{\Gamma_1}$ and $\tilde{\mathbf{B}}^{\Gamma_2}$, $\tau$ is a threshold. Once the correspondence features from different input views are obtained, these unknown instances will be updated as the foreground instances that can be fed into their original class head.

**Consistency Loss.** After obtaining the corresponding activation features of different branches, a consistency loss is utilized to ensure the consistency of the corresponding features as follows:

Table 3: Fine-tuning performance on Waymo benchmark (LEVEL_2 metric). Note that we only use a single checkpoint parameter to initialize all downstream baselines including SECOND, CenterPoint, PV-RCNN++. Semi denotes the semi-supervised method training on unlabeled ONCE split.

| Method | Paradigm | Data amount | L2 AP / APH | | | |
| --- | --- | --- | --- | --- | --- | --- |
| | | | Overall | Vehicle | Pedestrian | Cyclist |
| From scratch (SECOND) | - | 3% | 52.00 / 37.70 | 58.11 / 57.44 | 51.34 / 27.38 | 46.57 / 28.28 |
| From scratch (SECOND) | - | 20% | 60.62 / 56.86 | 64.26 / 63.73 | 59.72 / 50.38 | 57.87 / 56.48 |
| ProposalContrast (SECOND) [31] | SS-PT | 20% | 60.91 / 57.16 | 64.50 / 63.90 | 60.33 / 51.00 | 57.90 / 56.60 |
| BEV-MAE (SECOND) [13] | SS-PT | 20% | 61.03 / 57.30 | 64.42 / 63.87 | 59.97 / 50.65 | 58.69 / 57.39 |
| MeanTeacher (SECOND) [21] | Semi | 20% | 60.93 / 57.31 | 64.22 / 63.73 | 59.54 / 50.80 | 58.66 / 57.41 |
| Ours (SECOND) | AD-PT | 3% | 55.41 / 51.78 | 60.53 / 59.93 | 54.91 / 45.78 | 50.79 / 49.65 |
| Ours (SECOND) | AD-PT | 20% | **61.26 / 57.69** | **64.54 / 64.00** | 60.25 / **51.21** | **59.00** / 57.86 |
| From scratch (CenterPoint) | - | 3% | 59.00 / 56.29 | 57.12 / 56.57 | 58.66 / 52.44 | 61.24 / 59.89 |
| From scratch (CenterPoint) | - | 20% | 66.47 / 64.01 | 64.91 / 64.42 | 66.03 / 60.34 | 68.49 / 67.28 |
| GCC-3D (CenterPoint) [12] | SS-PT | 20% | 65.29 / 62.79 | 63.97 / 63.47 | 64.23 / 58.47 | 67.68 / 66.44 |
| ProposalContrast (CenterPoint) [31] | SS-PT | 20% | 66.67 / 64.20 | 65.22 / 64.80 | 66.40 / 60.49 | 68.48 / 67.38 |
| BEV-MAE (CenterPoint) [13] | SS-PT | 20% | 66.92 / 64.45 | 64.78 / 64.29 | 66.25 / 60.53 | **69.73 / 68.52** |
| MeanTeacher (CenterPoint) [21] | Semi | 20% | 66.66 / 64.23 | 64.94 / 64.43 | 66.35 / 60.61 | 68.69 / 67.65 |
| Ours (CenterPoint) | AD-PT | 3% | 61.21 / 58.46 | 60.35 / 59.79 | 60.57 / 54.02 | 62.73 / 61.57 |
| Ours (CenterPoint) | AD-PT | 20% | **67.17 / 64.65** | **65.33 / 64.83** | **67.16 / 61.20** | 69.39 / 68.25 |
| From scratch (PV-RCNN++) | - | 3% | 63.81 / 61.10 | 64.42 / 63.93 | 64.33 / 57.79 | 62.69 / 61.59 |
| From scratch (PV-RCNN++) | - | 20% | 69.97 / 67.58 | 69.18 / 68.75 | 70.88 / 65.21 | 69.84 / 68.77 |
| ProposalContrast (PV-RCNN++) [31] | SS-PT | 20% | 70.30 / 67.78 | 69.45 / 69.00 | 71.42 / 65.68 | 70.04 / 69.05 |
| BEV-MAE (PV-RCNN++) [13] | SS-PT | 20% | 70.54 / 68.11 | 69.53 / 69.07 | 71.50 / 65.69 | 70.60 / 69.56 |
| MeanTeacher (PV-RCNN++) [21] | Semi | 20% | 70.62 / 68.14 | 69.21 / 68.81 | 71.96 / 66.42 | 70.17 / 69.21 |
| Ours (PV-RCNN++) | AD-PT | 3% | 68.33 / 65.69 | 68.17 / 67.70 | 68.82 / 62.39 | 68.00 / 67.00 |
| Ours (PV-RCNN++) | AD-PT | 20% | **71.55 / 69.23** | **70.62 / 70.19** | **72.36 / 66.82** | **71.69 / 70.70** |

$$\mathcal{L}_{consist} = \frac{1}{BK} \sum_{i=1}^{B} \sum_{j=1}^{K} (\hat{f}_j^{\Gamma_1} - \hat{f}_j^{\Gamma_2})^2, \tag{4}$$

where $B$ is the batch size and $K$ is the number of the corresponding activation features.

**Overall Function.** The overall loss function can be formulated as:

$$\mathcal{L}_{total} = \mathcal{L}_{cls} + \mathcal{L}_{reg} + \mathcal{L}_{consist}, \tag{5}$$

where $\mathcal{L}_{cls}$ and $\mathcal{L}_{reg}$ represent classification loss and regression loss of the dense head, respectively, and $\mathcal{L}_{consist}$ is the activation consistent loss as shown in Eq. 4.

## 4 Experiments

### 4.1 Experimental Setup

**Pre-training Dataset.** ONCE [14] is a large-scale dataset collected in many scenes and weather conditions. ONCE contains 581 sequences composed of 20 labeled (∼19k frames) and 561 unlabeled sequences(∼1M frames). The labeled set divides into a train set with 6 sequences (∼5K samples), a validation set with 4 sequences (∼3k frames), and a test set with 10 sequences (∼8k frames). We merge Car, Bus, Truck into a unified category (*i.e.*, Vehicle) when performing the pre-training. Our main results are based on ONCE small split and use a larger split to verify the pre-training scalability.

**Description of Downstream Datasets.** *1) Waymo Open Dataset* [20] contains ∼150k frames. *2) nuScenes Dataset* [1] provides point cloud data from a 32-beam LiDAR consisting of 28130 training samples and 6019 validation samples. *3) KITTI Dataset* [5] includes 7481 training samples and is divided into a train set with 3712 samples and a validation set with 3769 samples.

**Description of Selected Baselines.** In this paper, we compare our method with several baseline methods including both Self-Supervised Pre-Training (SS-PT) and semi-supervised learning methods. **SS-PT methods**: as mentioned in Sec. 2.2, We mainly compare with contrastive learning-based (*i.e.*, GCC-3D [12], ProposalContrast [31]) and MAE-based SS-PT methods (*i.e.*, Voxel-MAE [6], BEV-MAE [13]). All these methods are pre-trained and fine-tuned on the same dataset. **Semi-supervised learning methods**: to verify the effectiveness of our method under the same experimental setting, we also compare with commonly-used semi-supervised technique (*i.e.*, MeanTeacher [21]).

Table 4: Fine-tuning performance on nuScenes benchmark. C.P. denotes that CenterPoint is employed as the baseline detector and D.A. represents the Data Amount. We fine-tune on 5% and 100% data for 20 epochs. Compared with other works [12, 13], our pre-training process is performed on a unified dataset rather than nuScenes.

| Method | Setting | D.A. | mAP | NDS | Car | Truck | CV. | Bus | Trailer | Barrier | Motor. | Bicycle | Ped. | TC. |
|---|---|---|---|---|---|---|---|---|---|---|---|---|---|---|
| From scratch (SECOND) | - | 5% | 29.24 | 39.74 | 67.69 | 33.02 | 7.15 | 45.91 | 17.67 | 25.23 | 11.92 | 0.00 | 53.00 | 30.74 |
| From scratch (SECOND) | - | 100% | 50.59 | 62.29 | - | - | - | - | - | - | - | - | - | - |
| Ours (SECOND) | AD-PT | 5% | 37.69 | 47.95 | 74.89 | 41.82 | 12.05 | 54.77 | 28.91 | 34.41 | 23.63 | 3.19 | 63.61 | 39.54 |
| Ours (SECOND) | AD-PT | 100% | **52.23** | **63.04** | 83.12 | 52.86 | 15.24 | 68.58 | 37.54 | 59.48 | 46.01 | 20.44 | 78.96 | 60.05 |
| From scratch (C.P.) | - | 5% | 42.68 | 50.41 | 77.82 | 43.61 | 10.65 | 44.01 | 18.71 | 52.95 | 36.26 | 16.76 | 67.62 | 54.52 |
| From scratch (C.P.) | - | 100% | 56.2 | 64.5 | 84.8 | 53.9 | 16.8 | 67.0 | 35.9 | 64.8 | 55.8 | 36.4 | 83.1 | 63.4 |
| GCC-3D (C.P.) [12] | SS-PT | 100% | **57.3** | 65.0 | **85.0** | **54.7** | **17.6** | 67.2 | 35.7 | 65.0 | 56.2 | 36.0 | 82.9 | 63.7 |
| BEV-MAE (C.P.) [13] | SS-PT | 100% | 57.2 | 65.1 | 84.9 | 54.9 | 16.5 | 67.2 | 35.9 | **65.2** | 56.0 | 36.2 | 83.2 | 63.5 |
| Ours (C.P.) | AD-PT | 5% | 44.99 | 52.99 | 78.90 | 43.82 | 11.13 | 55.16 | 21.22 | 55.10 | 39.03 | 17.76 | 72.28 | 55.43 |
| Ours (C.P.) | AD-PT | 100% | 57.17 | **65.48** | 84.86 | 54.37 | 16.09 | **67.34** | **36.06** | 64.31 | **58.50** | **40.58** | **83.53** | **66.05** |

Table 5: Fine-tuning performance ($AP_{3D}$) on KITTI benchmark.

| Method | Setting | Data amount | mAP (Mod.) | Car | | | Pedestrian | | | Cyclist | | |
|---|---|---|---|---|---|---|---|---|---|---|---|---|
| | | | | Easy | Mod. | Hard | Easy | Mod. | Hard | Easy | Mod. | Hard |
| From scratch (SECOND) | - | 20% | 61.70 | 89.78 | 78.83 | 76.21 | 52.08 | 47.23 | 43.37 | 76.35 | 59.06 | 55.24 |
| From scratch (SECOND) | - | 100% | 66.70 | 89.63 | 80.78 | 78.21 | 58.05 | 52.61 | 48.24 | 84.25 | 66.71 | 62.50 |
| Ours (SECOND) | AD-PT | 20% | 65.95 | 90.23 | 80.70 | 78.29 | 55.63 | 49.67 | 45.12 | 83.78 | 67.50 | 63.40 |
| Ours (SECOND) | AD-PT | 100% | **67.58** | **90.36** | **81.39** | **78.41** | **58.30** | **53.58** | **48.72** | **86.04** | **67.78** | **63.95** |
| From scratch (PV-RCNN) | - | 20% | 66.71 | 91.81 | 82.52 | 80.11 | 58.78 | 53.33 | 47.61 | 86.74 | 64.28 | 59.53 |
| ProposalContrast (PV-RCNN) [31] | SS-PT | 20% | 68.13 | 91.96 | 82.65 | 80.15 | 62.58 | 55.05 | 50.06 | 88.58 | 66.68 | 62.32 |
| From scratch (PV-RCNN) | - | 100% | 70.57 | - | 84.50 | - | - | 57.06 | - | - | 70.14 | - |
| GCC-3D (PV-RCNN) [12] | SS-PT | 100% | 71.26 | - | - | - | - | - | - | - | - | - |
| STRL (PV-RCNN) [7] | SS-PT | 100% | 71.46 | - | 84.70 | - | - | 57.80 | - | - | 71.88 | - |
| PointContrast (PV-RCNN) [25] | SS-PT | 100% | 71.55 | 91.40 | 84.18 | 82.25 | 65.73 | 57.74 | 52.46 | 91.47 | 72.72 | 67.95 |
| ProposalContrast (PV-RCNN) [31] | SS-PT | 100% | 72.92 | **92.45** | 84.72 | 82.47 | 68.43 | 60.36 | 55.01 | **92.77** | **73.69** | **69.51** |
| Ours (PV-RCNN) | AD-PT | 20% | 69.43 | 92.18 | 82.75 | 82.12 | 65.50 | 57.59 | 51.84 | 84.15 | 67.96 | 64.73 |
| Ours (PV-RCNN) | AD-PT | 100% | **73.01** | 91.96 | **84.75** | **82.53** | **68.87** | **60.79** | **55.42** | 91.81 | 73.49 | 69.21 |

**Implementation Details.** We fine-tune the model on several different detectors including SEC-OND [27], CenterPoint [32] and PV-RCNN++ [19]. Note that the transformer-based detectors are not used, since we consider directly applying the pre-trained backbone parameters to ***the most commonly used downstream baselines***. For different views generation, we consider 3 types of data augmentation methods, including random rotation ([$-180°$, $180°$]), random scaling ([0.7, 1.2]), and random flipping along X-axis and Y-axis. We use Adam optimizer with one-cycle learning rate schedule and the maximum learning rate is set to 0.003. We pre-train for 30 epochs on ONCE small split and large split using 8 NVIDIA Tesla A100 GPUs. The number of selected RoI features is set to 256 and the cross-view matching threshold $\tau$ is set to 0.3. Our code is based on 3DTrans [22].

**Evaluation Metric.** We use dataset-specific evaluation metrics to evaluate fine-tuning performance on each downstream dataset. ***For Waymo***, average precision (AP) and average precision with heading (APH) are utilized for three classes (*i.e.*, Vehicle, Pedestrian, Cyclist). Following [13, 31], we mainly focus on the more difficult LEVEL_2 metric. ***For nuScenes***, we report mean average precision (mAP) and NuScenes Detection Score (NDS). ***For KITTI***, we use mean average precision (mAP) with 40 recall to evaluate the detection performance and report $AP_{3D}$ results.

Table 6: Ablation study on data preparation.

| Method | Enhancement | Waymo L2 AP/APH | | | | nuScenes | |
|---|---|---|---|---|---|---|---|
| | | Overall | Vehicle | Pedestrian | Cyclist | mAP | NDS |
| Baseline | None | 67.12 / 64.55 | 67.45 / 66.97 | 67.74 / 61.15 | 66.19 / 65.24 | 36.26 | 45.04 |
| Baseline+re-scaling | Object-size | 67.39 / 64.68 | 67.52 / 67.03 | 67.82 / 61.24 | 66.83 / 65.79 | 39.72 | 49.93 |
| Baseline+re-sampling | LiDAR-beam | 67.37 / 64.70 | 67.70 / 67.21 | 68.21 / 61.71 | 66.15 / 65.18 | 41.35 | 51.03 |
| Baseline+re-scaling+re-sampling | Both | **67.77 / 65.09** | **68.01 / 67.61** | **68.32 / 61.69** | **66.99 / 65.98** | **43.11** | **52.41** |

## 4.2 Main Results

**Results on Waymo.** Results on Waymo validation set are shown in Tab. 3. We first compare the proposed method with previous SS-PT methods. It can be seen that all three detectors achieve the best results using AD-PT initialization, surpassing previous SS-PT methods even using a smaller pre-training dataset (*i.e.*, $\sim$100k frames on ONCE small split). For example, the improvement achieved by PV-RCNN++ is $1.58\%$ / $1.65\%$ in terms of L2 AP / APH. Note that the compared SS-PT

Table 7: The impact of pseudo-labeling methods on downstream datasets.

| Pseudo-labeling Method | ONCE | Waymo L2 AP/APH | | | | nuScenes | |
| --- | --- | --- | --- | --- | --- | --- | --- |
| | Overall | Overall | Vehicle | Pedestrian | Cyclist | mAP | NDS |
| SECOND (Low Performance) | 57.10 | 65.96 / 63.29 | 65.95 / 65.46 | 66.87 / 60.36 | 65.07 / 64.06 | 41.49 | 50.82 |
| CenterPoint (Middle Performance) | 60.84 | 66.79 / 64.10 | 67.09 / 66.60 | 67.79 / 61.16 | 65.51 / 64.55 | 41.91 | 51.64 |
| Ours (High Performance) | 69.90 | **67.77 / 65.09** | **68.01 / 67.61** | **68.32 / 61.69** | **66.99 / 65.98** | **43.11** | **52.41** |

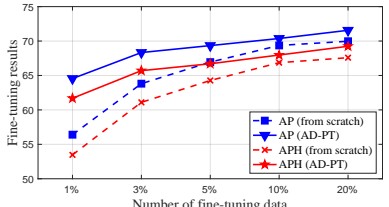

Figure 6: Different budgets

Table 8: The performance scalability using KITTI and ONCE.

| Pre-training dataset | Waymo L2 AP/APH | | | |
| --- | --- | --- | --- | --- |
| | Overall | Vehicle | Pedestrian | Cyclist |
| KITTI ($\sim$4k) | 64.28 / 63.16 | 64.73 / 64.19 | 64.43 / 57.30 | 63.69 / 62.60 |
| ONCE ($\sim$4k) | 64.28 / 61.36 | 66.11 / 65.64 | 66.26 / 59.51 | 65.39 / 64.35 |
| ONCE ($\sim$10k) | 66.94 / 64.24 | 67.41 / 66.91 | 67.97 / 61.39 | 65.45 / 64.43 |
| ONCE ($\sim$100k) | 68.33 / 65.69 | 68.17 / 67.70 | 68.82 / 62.39 | 68.00 / 67.00 |
| ONCE ($\sim$500k) | **69.04 / 66.52** | **68.69 / 68.23** | **69.81 / 63.74** | **68.61 / 67.60** |

methods are pre-trained on Waymo 100% unlabeled train set ($\sim$150k frames), which has a smaller domain gap with fine-tuning data. Further, to verify the effectiveness of fine-tuning with a small number of samples, we conduct experiments of fine-tuning on 3% Waymo train set ($\sim$5K frames). We can observe that, with the help of pre-trained prior knowledge, fine-tuning with a small amount of data can achieve much better performance than training from scratch (*e.g.*, 3.41% / 14.08% in L2 using SECOND as baseline). In addition, to ensure the fairness of the experiments, we compare our method with semi-supervised learning methods which are identical to our pre-training setup.

**Results on nuScenes.** Due to the huge domain discrepancies, few works can transfer the knowledge obtained by pre-training on other datasets to nuScenes dataset. Thanks to constructing a unified large-scale dataset and considering taxonomic differences in the pre-training stage, our methods can also significantly improve the performance on nuScenes. As shown in Tab. 4, AD-PT improves the performance of training from scratch by 0.93% and 0.98% in mAP and NDS.

**Results on KITTI.** As shown in Tab. 5, we further fine-tune on a relatively small dataset (*i.e.*, KITTI). Note that previous methods often use models pre-trained on Waymo as an initialization since the domain gap is quite small. It can be observed that using AD-PT initialization can further improve the accuracy when fine-tuning on both 20% and 100% KITTI training data under different detectors. For instance, the $AP_{3D}$ in moderate level can improve 2.72% and 1.75% when fine-tuning on 20% and 100% KITTI data using PV-RCNN++ as the baseline detector.

### 4.3 Insight Analyses

In this part, we further discuss the 3D pre-training. Note that in ablation studies, we fine-tune on 3% Waymo data and 5% nuScenes data under the PV-RCNN++ and CenterPoint baseline setting.

#### 4.3.1 Insight Analyses on the Data Preparation

**Discussion on Diversity-based Pre-training Processor.** From Tab. 6, we observe that both beam re-sampling and object re-scaling during the pre-training stage can improve the performance for other downstream datasets. For example, the overall performance can be improved by 0.65% / 0.54% on Waymo and 7.37% on nuScenes. Note that when pre-training without our data processor, the fine-tuning performance on nuScenes will drop sharply compared with training from scratch, due to the large domain discrepancy. Since our constructed unified dataset can cover diversified data distributions, the backbone can learn more general representations.

**Pseudo-labeling Performance.** Tab. 7 indicates the impact of pseudo-labeling operation on the downstream perception performance. We use three types of pseudo-labeling methods to observe the low, middle, and high performance on ONCE. We find that the performance of fine-tuning is positively correlated with the accuracy of pseudo-labeling on ONCE. This is mainly due to that pseudo labels with relatively high accuracy can guide the backbone to activate more precise foreground features and meanwhile suppress background regions.

**Scalability.** To verify the scaling ability of the AD-PT paradigm, we conduct experiments in Tab 8 to show the Waymo performance initialized by different pre-trained checkpoints, which are obtained using pre-training datasets with different scales. Please refer to Appendix for more results.

### 4.3.2  Insight Analyses on the Unified Representations Learning

**Discussion on Unknown-aware Instance Learning Head.** It can be seen from Tab. 9 that, Unknown-aware Instance Learning (UIL) head and Consistency Loss (CL) can further boost the perception accuracy on multiple datasets. It can be observed that the gains are larger on the Pedestrian and Cyclist categories. The reason is that the UIL head can better capture downstream-sensitive instances, which are hard to be detected during the pseudo-labeling **pre-training** process.

Table 9: Ablation study on the designed UIL and CL.

| Method | Waymo L2 AP/APH | | | | nuScenes | |
|---|---|---|---|---|---|---|
| | Overall | Vehicle | Pedestrian | Cyclist | mAP | NDS |
| Baseline | 67.77 / 65.09 | 68.01 / 67.61 | 68.32 / 61.69 | 66.99 / 65.98 | 43.11 | 52.41 |
| Baseline+UIL | 67.97 / 65.35 | 67.99 / 67.58 | 68.62 / 62.12 | 67.32 / 66.35 | 43.92 | 52.65 |
| Baseline+UIL+CL | **68.33 / 65.69** | **68.17 / 67.70** | **68.82 / 62.39** | **68.00 / 67.00** | **44.99** | **52.99** |

**Training-efficient Method.** In the main results, we verify that our methods can bring performance gains under different amounts of fine-tuning data (*e.g.*, 1%, 5%, 10% budgets). As shown in Fig. 6, results demonstrate that our method can consistently improve performance under different budgets.

### 4.3.3  Insight Analysis on Different Types of Baseline Detectors

To verify the generalization of our proposed method, we further conduct experiments on different types of 3D object detection backbones (*i.e.*, pillar-based and point-based). As shown in Tab. 10 and Tab. 11, the performance of multiple types of detectors can improve when initialized by AD-PT pre-trained checkpoints which further shows that AD-PT is a general pre-training pipeline that can be used on various types of 3D detectors.

Table 10: Ablation study on the pillar-based backbone. We conduct experiments on Waymo dataset.

| Method | Data amount | Waymo L2 AP/APH | | | |
|---|---|---|---|---|---|
| | | Overall | Vehicle | Pedestrian | Cyclist |
| From scratch (PointPillar) | 20% | 48.56 / 39.30 | 54.28 / 53.51 | 47.11 / 25.50 | 44.29 / 38.89 |
| AD-PT (PointPillar) | 20% | **52.01 / 43.99** | **58.51 / 57.85** | **50.22 / 32.52** | **47.31 / 41.59** |
| From scratch (PointPillar) | 100% | 57.85 / 50.69 | 62.18 / 61.64 | 58.18 / 40.64 | 53.18 / 49.80 |
| AD-PT (PointPillar) | 100% | **59.71 / 53.49** | **64.10 / 63.54** | **59.00 / 43.13** | **56.04 / 53.80** |

Table 11: Ablation study on the point-based backbone. We conduct experiments on KITTI dataset.

| Method | Data amount | mAP | Car | | | Pedestrian | | | Cyclist | | |
|---|---|---|---|---|---|---|---|---|---|---|---|
| | | (Mod.) | Easy | Mod. | Hard | Easy | Mod. | Hard | Easy | Mod. | Hard |
| From scratch (PointRCNN) | 20% | 64.12 | - | 75.30 | - | - | 52.52 | - | - | 69.55 | - |
| AD-PT (PointRCNN) | 20% | **67.67** | 88.75 | 77.20 | 75.18 | 64.58 | 54.16 | 48.24 | 84.25 | 71.86 | 62.50 |
| From scratch (PointRCNN) | 100% | 68.40 | - | 78.70 | - | - | 54.41 | - | - | 72.11 | - |
| AD-PT (PointRCNN) | 100% | **70.47** | 90.90 | 80.25 | 78.05 | 64.80 | 57.13 | 50.37 | 92.45 | 74.04 | 69.45 |

## 5  Conclusion

In this work, we have proposed the AD-PT paradigm, aiming to pre-train on a unified dataset and transfer the pre-trained checkpoint to multiple downstream datasets. We comprehensively verify the generalization ability of the built unified dataset and the proposed method by testing the pre-trained model on different downstream datasets including Waymo, nuScenes, and KITTI, and different 3D detectors including PV-RCNN, PV-RCNN++, CenterPoint, and SECOND.

## 6  Limitation

Although the AD-PT pre-trained backbone can improve the performance on multiple downstream datasets, it needs to be verified in more actual road scenarios. Meanwhile, training a backbone with more generalization capabilities through data from different sensors is also a future direction.

## Acknowledgement

This work is supported by National Natural Science Foundation of China (No. 62071127), National Key Research and Development Program of China (No. 2022ZD0160100), Shanghai Natural Science Foundation (No. 23ZR1402900) and Shanghai Artificial Intelligence Laboratory, in part by Science and Technology Commission of Shanghai Municipality under Grant 22DZ1100102, in part by National Key R&D Program of China under Grant No. 2022ZD0160104.

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

# Supplementary Material

In this supplementary material, we provide more details and experimental results not included in our main text.

**Outlines:**

# A    More Details about Large-scale Pre-training Dataset Preparation.

In this section, we give some preliminary experimental results and analysis on large-scale pre-training dataset preparation.

## A.1    Preliminary Experiments on Class-aware Pseudo Label Generator

As mentioned in Sec. 3.2.1 in our submission, we explore how to improve the performance on ONCE. We first analyze the results in the ONCE benchmark and find that CenterPoint reaches the SOTA performance on pedestrian and cyclist while PV-RCNN achieves the best performance on vehicle. To use a stronger baseline to further improve the performance, we conduct experiments using PV-RCNN++ as the baseline detector. As shown in Tab. 12, PV-RCNN++ with center head can not obtain a satisfactory performance on ONCE while PV-RCNN++ with anchor head can achieve better accuracy on vehicle and pedestrian.

Further, to obtain more accurate pseudo labels, we use a semi-supervised learning method to further improve the performance as shown in Tab. 13. Finally, we individually train pedestrian using CenterPoint and other classes using PV-RCNN++.

Table 12: Effects of using different heads on PV-RCNN++. We report mAP using the ONCE evaluation metric.

| Detector | Head Choice | Vehicle | Pedestrian | Cyclist |
|---|---|---|---|---|
| PV-RCNN++ | Center Head | 71.61 | **45.27** | 61.15 |
| PV-RCNN++ | Anchor Head | **81.72** | 43.86 | **66.17** |

## A.2    Analysis on Pseudo Label Threshold on Different Classes

Fig. 7 shows the precision under different IoU thresholds. The precision can be calculated by $\text{Precision} = \text{TP}/(\text{FP} + \text{TP})$, where FP and TP denote false positive and true positive, respectively. We can observe that when IoU thresholds are more than 0.8, 0.7, 0.7 for vehicle, pedestrian and cyclist, the number of TP instances is significantly more than that of FP instances.

Table 13: Effects of using MeanTeacher. We report mAP using the ONCE evaluation metric.

| Detector | MeanTeacher | Vehicle | Pedestrian | Cyclist |
|----------|:-----------:|:-------:|:----------:|:-------:|
| CenterPoint | ✗ | - | 46.22 | - |
| CenterPoint | ✓ | - | **56.01** | - |
| PV-RCNN++ | ✗ | 81.72 | - | 66.17 |
| PV-RCNN++ | ✓ | **82.50** | - | **71.19** |

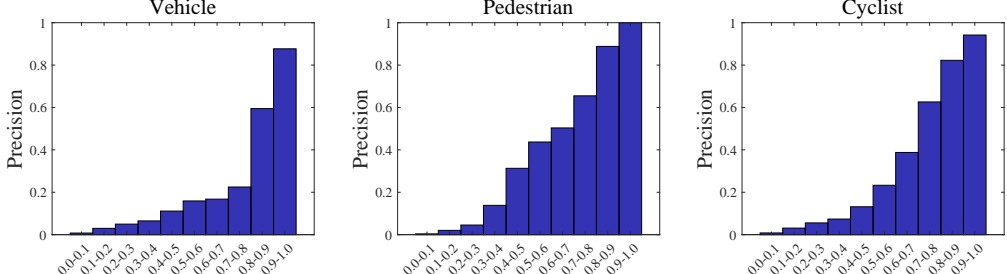

Figure 7: The Precision under different IoU thresholds.

The visualization of the pseudo label results under different thresholds in Fig. 8, we can see that some FP pseudo labels will be annotated when setting low thresholds, while some TP instances can not be annotated when the thresholds are relatively high. To more intuitively see the impact of the threshold on pseudo labeling, we use the model to annotate the samples of the ONCE validation set for comparison with ground-truths.

## A.3 Visualization Results of Pseudo Labels

Fig. 9 shows the visualization results of our final pseudo label results.

## A.4 Details of Object Re-scaling

In detail, given a bounding box $b = (c_x, c_y, c_z, l, w, h, \theta_h)$ and point clouds $(p_i^x, p_i^y, p_i^z)$ within it, where $(c_x, c_y, c_z)$, $(l, w, h)$ and $\theta_h$ denote the center, size and heading angle of the bounding box. We first transform points into the local coordinate with the following formula:

$$
(p_i^l, p_i^w, p_i^h) = (p_i^x - c_x, p_i^y - c_y, p_i^z - c_z) \cdot R,
$$
$$
R = \begin{bmatrix} \cos\theta_h & -\sin\theta_h & 0 \\ \sin\theta_h & \cos\theta_h & 0 \\ 0 & 0 & 1 \end{bmatrix}, \tag{6}
$$

where $\cdot$ is matrix multiplication. Then, to derive the scaled object, the point coordinates inside the box and the bounding box size are scaled to be $\alpha(p_i^l, p_i^w, p_i^h)$ and $\alpha(l, w, h)$, where $\alpha$ is the scaling factor. Finally, the points inside the scaled box are transformed back to the ego-car coordinate system and shifted to the center $(c_x, c_y, c_z)$ as

$$
\tilde{p}_i = \alpha(p_i^l, p_i^w, p_i^h) \cdot R^T + (c_x, c_y, c_z). \tag{7}
$$

## A.5 Taxonomy difference between different datasets

As shown in Tab. 14, there exists a huge taxonomy difference between some fine-tuning datasets and the pre-training dataset. As a result, some foreground instances may be regarded as background if only using pseudo label as supervision.

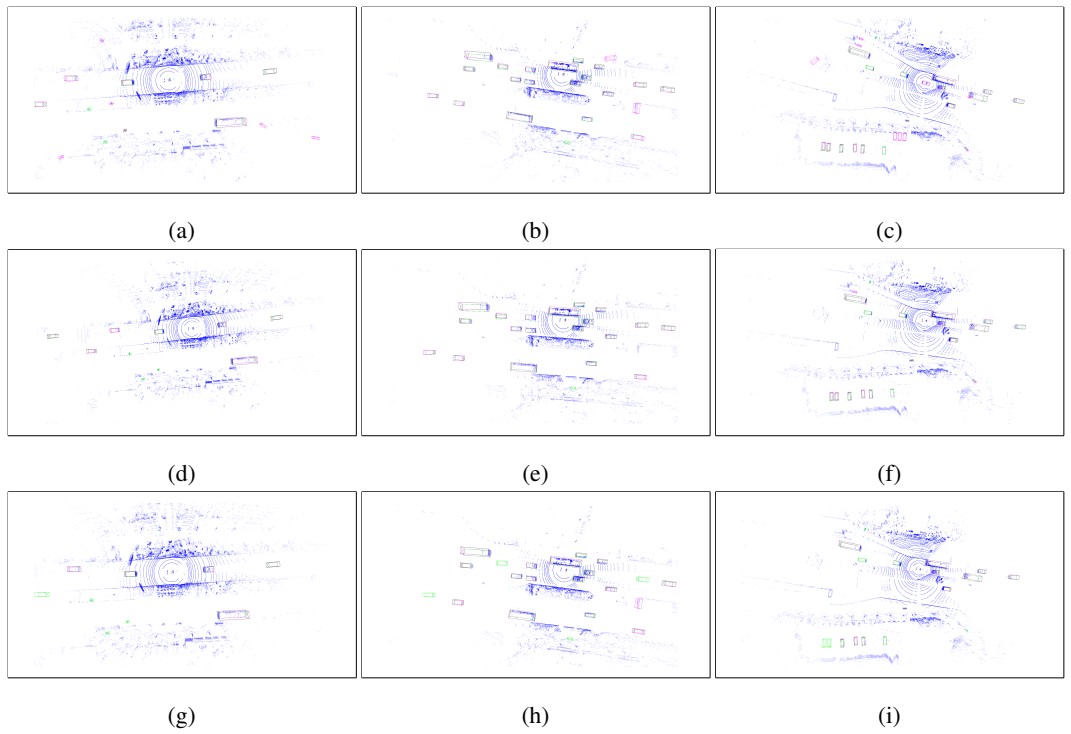

Figure 8: Visualization results under different pseudo label thresholds. (a-c): annotations with low thresholds (*i.e.*, 0.6, 0.5, 0.5 for vehicle, pedestrian and cyclist, respectively). (d-f): the thresholds used in our methods. (g-i): high thresholds (*i.e.*, 0.9, 0.8, 0.8 for vehicle, pedestrian and cyclist, respectively). The green and red bounding boxes represent ground-truths and detector predictions, respectively.

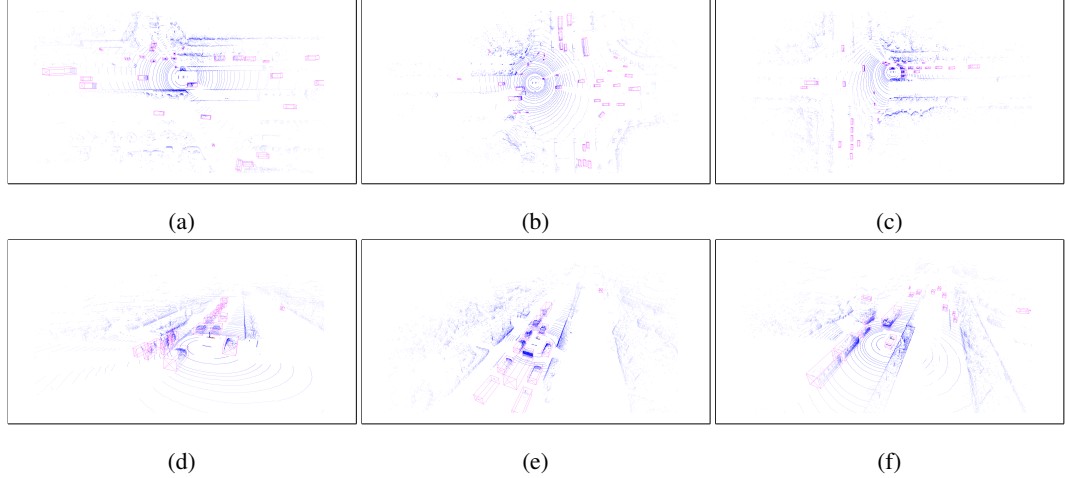

Figure 9: Pseudo-labeled annotation results on unlabeled set.

## B   Detailed Dataset Description and Evaluation Metrics

### B.1   Dataset Description

**ONCE dataset.**   ONCE dataset [14] is a large-scale dataset that is built to encourage the exploration of self-supervised and semi-supervised learning in the autonomous driving scenario. ONCE is collected by a 40-beam LiDAR in multiple cities in China and contains diverse weather conditions

Table 14: Taxonomy difference between different datasets.

| Dataset | classes |
|---|---|
| ONCE (Pre-train) | Car, Truck, Bus, Pedestrian, Cyclist |
| Waymo (Fine-tune) | Vehicle, Pedestrian, Cyclist |
| nuScenes (Fine-tune) | Car, Truck, Construction vehicle, Bus, Trailer, Barrier, Motorcycle, Bicycle, Pedestrian, Traffic cone |
| KITTI (Fine-tune) | Car, Pedestrian, Cyclist |

(*e.g.*, sunny, cloudy, rainy), traffic conditions, time periods (*e.g.*, morning, noon, afternoon, night) and areas (*e.g.*, downtown, suburbs, highway, tunnel, bridge).

**Waymo Open Dataset.** Waymo Open Dataset [20] is a widely-used large-scale autonomous driving dataset that is composed of 1000 sequences and divided into a train set with 798 sequences (~150k samples) and a validation set with 202 sequences (~40k samples). The Waymo dataset is gathered in the USA by a 64-beam LiDAR and 4 200-beam short-range LiDAR with annotations in full $360°$. We use the 1.0 version of Waymo Open Dataset.

**nuScenes Dataset.** NuScnenes dataset [1] provides point cloud data from 32-beam LiDAR collected from Singapore and Boston, USA. It consists of 28130 training samples and 6019 validation samples. The data is obtained during different times in the day, different weather conditions and a diverse set of locations (*e.g.*, urban, residential, nature and industrial).

**KITTI Dataset.** KITTI dataset [5] is a common-used autonomous driving dataset that contains 7481 training samples and is divided into a train set with 3712 samples and a validation set with 3769 samples. The point cloud data is collected by a 64-beam LiDAR in Germany. KITTI dataset only provides the annotations for the objects within the field of view of the front RGB camera.

## B.2 Evaluation Metrics

**ONCE evaluation metric.** Following ONCE official evaluation metric, we merge the car, bus and truck class into a super-class (*i.e.*, vehicle). $AP_{3D}^{Ori}$ is used to evaluate the performance of the ONCE dataset, which can be obtained by the following formula:

$$AP_{3D}^{Ori} = 100 \int_0^1 max\{p(r'|r' \geq r)\}dr, \tag{8}$$

where $r$ is recall rates from 0.02 to 1.00 at step 0.02 and $p(r)$ denotes the precision-recall curve. Mean average precision (mAP) is the average of the scores of the three categories. The Intersection over Union (IoU) thresholds are set to 0.7, 0.3 and 0.5 for vehicle, pedestrian and cyclist, respectively.

**Waymo evaluation metric.** Two difficulty levels (*i.e.*, LEVEL 1 and LEVEL 2) are utilized to evaluate the detection accuracy of Waymo dataset and we mainly focus on more difficult L2 performance. Among each difficulty level, we report AP and APH which can be formulated as:

$$AP = 100 \int_0^1 max\{p(r')|r' \geq r\}dr, \quad AP = 100 \int_0^1 max\{h(r')|r' \geq r\}dr, \tag{9}$$

where the different between $h(r)$ and $p(r)$ is $h(r)$ is weighted by the accuracy of heading accuracy.

**nuScenes evaluation metric.** Following the official NuScenes Evaluation Metric, we report mAP and nuScenes detection score (NDS). AP is defined as matches by thresholding the 2D center distance d on the ground plane and the mAP can be calculated by:

$$mAP = \frac{1}{\mathbb{C}} \frac{1}{\mathbb{D}} \sum_{c \in \mathbb{C}} \sum_{d \in \mathbb{D}} AP, \tag{10}$$

where $\mathbb{C}$ is the set of classes and $\mathbb{D}$ is the set of thresholds (*i.e.*, {0.5,1,2,4}). We mainly focus on 10 classes. NDS is the weighted of mAP and five true positive metrics, including Average Translation

Error (ATE), Average Scale Error (ASE), Average Orientation Error (AOE), Average Velocity Error (AVE) and Average Attribute Error (AAE). The NDS can be formulated as:

$$mTP = \frac{1}{\mathbb{C}} \sum_{c \in \mathbb{C}} TP_c, \quad NDS = \frac{1}{10}[5mAP + \sum_{mTP \in \mathbb{TP}} (1 - min(1, mTP))], \qquad (11)$$

where $\mathbb{TP}$ is the set of true positive metrics.

**KITTI evaluation metric.** We report mAP with 40 recall positions to evaluate the detection performance and the 3D IoU thresholds is set to 0.7 for cars and 0.5 for pedestrians and cyclists.

## C  More Implementation Details

As shown in Tab. 15, we list some details about pre-training and fine-tuning datasets. Note that the voxel size of nuScenes is set to [0.1, 0.1, 0.2] following [13]. It can be seen that different datasets may have different dimensions of input features (*e.g.*, ONCE use 4 dimension features as input while Waymo and nuScenes use 5 dimension features) causing the input dimension of the first layer network to be different. We simply do not load the parameters of the first layer when this happens while fine-tuning. In the pre-training phase, we merge the pseudo-labeled data and a small amount of labeled data (*i.e.*, ONCE train set) as the pre-training dataset. In the fine-tuning phase, we fine-tune 30 epochs for Waymo, 20 epochs for nuScenes and 80 epochs for KITTI.

Table 15: Some implementation details about pre-training and fine-tuning datasets.

| Dataset | Point cloud range | voxel size | input features |
|---|---|---|---|
| ONCE (Pre-train) | [-75.2, -75.2, -5.0, 75.2, 75.2, 3.0] | [0.1, 0.1, 0.2] | [x, y, z, intensity] |
| Waymo (Fine-tune) | [-75.2, -75.2, -2.0, 75.2, 75.2, 4.0] | [0.1, 0.1, 0.15] | [x, y, z, intensity, elongation] |
| nuScenes (Fine-tune) | [-51.2, -51.2, -5.0, 51.2, 51.2, 3.0] | [0.1, 0.1, 0.2] | [x, y, z, intensity, timestamp] |
| KITTI (Fine-tune) | [0.0, -40.0, -3.0, 70.4, 40.0, 1.0] | [0.05, 0.05, 0.1] | [x, y, z, intensity] |

## D  More Experimental Results

### D.1  Ablation Studies on Unknown-aware Instance Learning Head

In this part, we conduct experiments to ablate the hyper-parameters in unknown-aware instance learning head (*i.e.*, the number $M$ of selected features and the distance threshold $\tau$).

Tab. 16 shows the results using different numbers of selected features in unknown-aware instance learning head when pre-training. When $M$ is small, some foreground instances with relatively low scores are ignored, while when $M$ is large, the matched background regions are increased. Considering these factors, we choose $M$ to be 256.

Tab. 17 shows the performance under different distance thresholds in Eq. 4 in the main submission. The number of matched features is relatively small when using a lower $\tau$, thus can not fully exploit the unknown foreground instances. When using a larger threshold, some mismatches may occur. Finally, we set $\tau$ to 0.3 as mentioned in our main submission.

Table 16: Ablation studies of the number $M$ of selected features.

| $M$ | Waymo L2 AP / APH | | | |
|---|---|---|---|---|
| | Overall | Vehicle | Pedestrian | Cyclist |
| 128 | 67.71 / 64.98 | 67.91 / 67.45 | 68.54 / 61.87 | 66.67 / 65.63 |
| 256 | **68.33 / 65.69** | **68.17 / 67.70** | **68.82 / 62.39** | **68.00 / 67.00** |
| 512 | 67.93 / 65.24 | 68.04 / 67.36 | 68.63 / 62.12 | 67.14 / 66.23 |

### D.2  More Results of Pre-training Scalability

In this section, we show more results to verify the pre-training scalability. We pre-train the model on the small, middle and large splits of the ONCE dataset and then fine-tune the model on 3% Waymo

Table 17: Ablation studies of the distance threshold $\tau$.

| $\tau$ | Waymo L2 AP/APH | | | |
| --- | --- | --- | --- | --- |
| | Overall | Vehicle | Pedestrian | Cyclist |
| 0.1 | 67.90 / 65.22 | 68.01 / 67.54 | 68.52 / 62.01 | 67.17 / 66.12 |
| 0.3 | **68.33 / 65.69** | **68.17 / 67.70** | **68.82 / 62.39** | **68.00 / 67.00** |
| 0.5 | 67.82 / 65.15 | 67.73 / 67.26 | 68.26 / 61.73 | 67.49 / 66.46 |

and 20% KITTI train data. As shown in Tab. 18, as the scale of the pre-training dataset and the diversity of scenarios increases, the performance of fine-tuning on the downstream dataset will also improve.

Table 18: The pre-training scalability. We use ONCE to pre-train and Waymo and KITTI to fine-tune.

| Pre-training dataset | Waymo L2 AP/APH | | | | KITTI Moderate mAP | | | |
| --- | --- | --- | --- | --- | --- | --- | --- | --- |
| | Overall | Vehicle | Pedestrian | Cyclist | Overall | Car | Pedestrian | Cyclist |
| ONCE (∼100k) | 68.33 / 65.69 | 68.17 / 67.70 | 68.82 / 62.39 | 68.00 / 67.00 | 69.43 | 82.75 | 57.59 | 67.96 |
| ONCE (∼500k) | 69.04 / 66.52 | 68.69 / 68.23 | 69.81 / 63.74 | 68.61 / 67.60 | 71.36 | 83.17 | 58.14 | 72.78 |
| ONCE (∼1M) | **69.63 / 67.08** | **69.03 / 68.57** | **70.54 / 64.34** | **69.33 / 68.33** | **72.37** | **83.47** | **59.84** | **73.81** |

## D.3 Results of fine-tuning on ONCE.

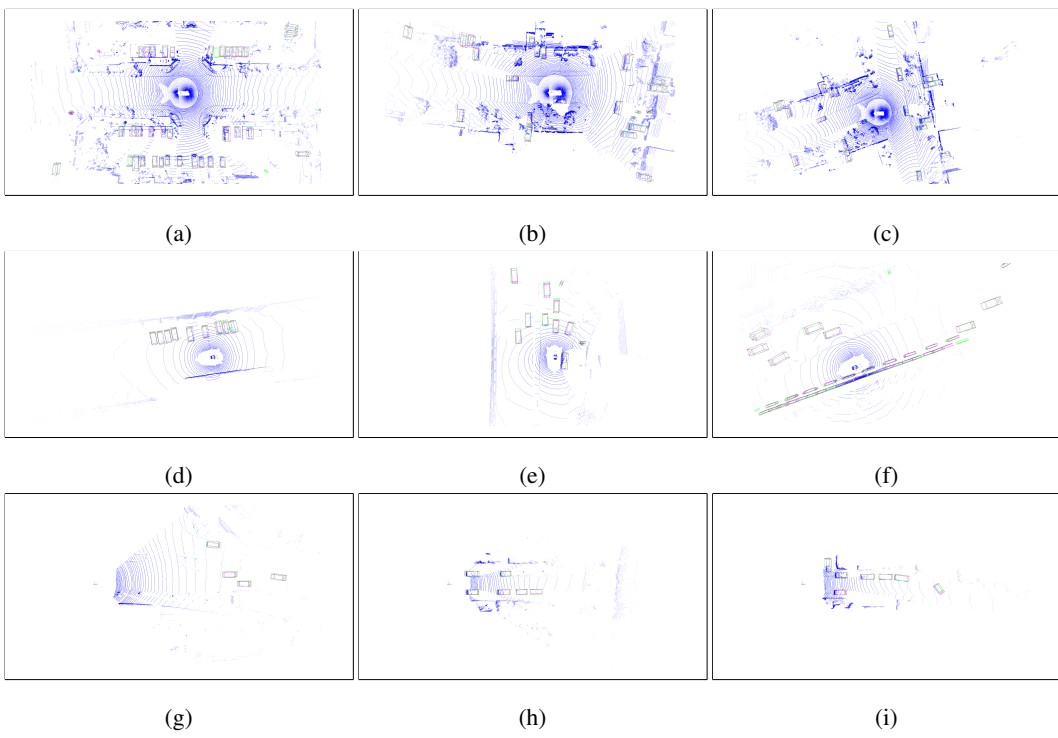

Figure 10: Visualization of fine-tuning results. We visualize the results of three downstream datasets. (a-c): results of Waymo. (d-f): results of nuScenes. (g-i): results of KITTI.
The green and red bounding boxes represent ground-truths and detector predictions, respectively.

In our main submission, we report the fine-tuning performance on multiple datasets which are different from the pre-training dataset. Here, we show some fine-tuning performance on ONCE. As shown in Tab. 19, the performance can be largely improved when the baseline detectors are initialized by AD-PT. For example, when using SECOND as the baseline detector, the overall performance can be improved from 56.47% to 64.10% (+7.63%). We use the ONCE train set to fine-tune the model.

Table 19: The fine-tuning performance on ONCE validation set.

| Init. | SECOND | | | | CenterPoint | | | |
|---|---|---|---|---|---|---|---|---|
| | Overall | 0-30m | 30-50m | >50m | Overall | 0-30m | 30-50m | >50m |
| Random Initialization | 56.47 | 65.94 | 51.05 | 36.44 | 64.94 | 74.52 | 59.47 | 44.28 |
| AD-PT Initialization | **64.10** | **74.34** | **57.69** | **41.23** | **67.73** | **76.48** | **61.85** | **46.29** |

## D.4 Visualization Results.

Fig. 10 shows the visualization results of three downstream datasets (*i.e.*, Waymo, nuScenes, KITTI).

