# OpenReview forum: "AD-PT: Autonomous Driving Pre-Training with Large-scale Point Cloud Dataset"
_NeurIPS.cc/2023/Conference — NeurIPS 2023 poster_

### Official Review · Reviewer_5kGB · 2023-07-01

**Soundness:** 3 good
**Presentation:** 3 good
**Contribution:** 3 good
**Rating:** 6
**Confidence:** 3

**Summary:**

This paper studies model pre-training for autonomous driving by utilizing a large-scale point cloud dataset, which largely facilitates the generalization performance of pre-trained models. To reconstruct such a dataset, scene- and instance-level distribution diversity are carefully enhanced. The overall pretraining is conducted in a semi-supervised manner, and its effectiveness is validated on several benchmark datasets with different baseline models.

**Strengths:**

1. Compared to the traditional pre-training setting for point cloud, the AD-PT pretraining setting is more practical since it is expected to learn a generalized pre-trained model for all downstream tasks.
2. Compared to existing pre-trained methods trained in a fully unsupervised manner, this paper adopts a semi-supervised strategy, which is somewhat novel.
3. The data diversity enhancement with re-sampling strategies is reasonable. So does the unknown aware instance learning head.

**Weaknesses:**

1. Although the AD-PT presented an improved performance on downstream tasks with limited training data (e.g., 20% data amount on Waymo), it is interesting to illustrate the results with more training samples in downstream tasks (e.g., 100% data amount on Waymo).

**Questions:**

Na

---

> ### Author Rebuttal · Authors · 2023-08-09
>
> Response to Reviewer 5kGB
>
> Thanks for your review and suggestions.
>
> **Q1: Although the AD-PT presented an improved performance on downstream tasks with limited training data (e.g., 20% data amount on Waymo), it is interesting to illustrate the results with more training samples in downstream tasks (e.g., 100% data amount on Waymo).**
>
> **A1**: For a fair comparison with the previous works like ProposalContrast and BEV-MAE, we conduct experiments on fine-tuning using 20% Waymo data in the main text. Here, we show the results fine-tuned on 100% Waymo data.
>
>   |Method|Data amount | Overall L2 AP / APH | Vehicle| Pedestrian | Cyclist|
>   | --- | --- | --- | --- | --- | --- |
>   |From scratch (PV-RCNN++) | 100%| 71.66 / 69.45 | 70.61 / 70.18 | 73.17 / 68.00 | 71.21 / 70.19 |
>   |AD-PT (PV-RCNN++) | 100% | 72.39 / 70.10 | 71.01 / 70.60 | 74.84 / 69.46 | 71.32 / 70.25 |
>
>   It can be seen that the performance can be consistently improved using more fine-tuning data.

---

### Official Review · Reviewer_nLKe · 2023-07-01

**Soundness:** 3 good
**Presentation:** 3 good
**Contribution:** 3 good
**Rating:** 5
**Confidence:** 3

**Summary:**

This paper focuses on the point-cloud pre-training problem in the autonomous driving scenarios. Particularly, the authors regard the pre-training task as a semi-supervised learning problem and generate pseudo-labels for massive unlabeled data with few labeled frames. In the pseudo-label generation phase, they proposed two specific designs for this task, including diversity-based pre-training data preparation and unknown-aware instance learning. The experiments conducted on the KITTI, nuScenes, and Waymo datasets show the effectiveness of the proposed method.

**Strengths:**

- The topic is meaningful. Different from the 2D vision task, the 3D detection community lacks effective pretraining methods and pre-trained backbones, and the strong pre-training backbones are helpful to the whole community.
- The authors provide a pre-trained paradigm, and the experiments show the effectiveness of the pre-trained model, especially when training samples are limited.
- Code will be publicly available. The authors provide source code for reproducibility.

**Weaknesses:**

- The work claims that it provides a general pre-trained backbone for the LiDAR-based 3D object detection task. However, different from 2D perception tasks which are dominated by a few kinds of backbones,  there are several popular detection pipelines with different backbone nets in this field. Some 3D detectors even use different input data representations, e.g. points, pillars, voxels, range images, etc. It is hard to build unified 3D pre-trained models for these works, and what this work provided is a voxel-based pre-trained backbone. Based on this, I think this work is over-claimed.

- The 3D-based backbone is highly customized. Take the voxel-based backbone as an example, we need to pre-define the voxel size, and I think the pre-trained voxel size used in pretraining is the same as that used in the downstream task. I would like to know what would happen if different voxel sizes were used for pre-training and downstream tasks. If we must re-pretrain each backbone with different resolutions, then the significance of this work is not very important.

- The improvement in performance is limited. I found that the pre-trained model has a relatively small impact on the final performance, especially in the full dataset setting. Considering the extra pre-training cost, performance improvement is limited, and I  want to know whether simply extending the training schedule can achieve comparable performance or not.

- Novelty is limited. This work regards the pre-training task as a semi-supervised learning problem with a pseudo-labeling scheme. However, using a few labeled data to generate pseudo-labels for LiDAR-based 3D detection has been investigated in [1], which achieves better performance.

[1] Pseudo-labeling for Scalable 3D Object Detection, https://arxiv.org/abs/2103.02093


**Questions:**

See Weaknesses.

**Limitations:**

See Weaknesses.

---

> ### Author Rebuttal · Authors · 2023-08-09
>
> **Q1: More results of using the points, pillars, voxel-based methods to demonstrate the generalization ability of AD-PT.**
>
> **A1**:
> - As mentioned by the reviewer, there are several different types of 3D object detection backbones, *e.g.* pillar-based, point-based, and voxel-based. We verify the generalization ability of AD-PT based on the voxel-based method, due to it being the most commonly-used backbone currently.
> - Furthermore, we conduct experiments on different types of backbones, *e.g.* pillar- and point-based. The results are shown in the following tables.
>   |Methods (PointPillar)|Data amount (Waymo)|Overall L2 AP/APH|Veh.|Ped.|Cyc.|
>   |-|-|-|-|-|-|
>   |Scratch|3%|48.56/39.30|54.28/53.51|47.11/25.50|44.29/38.89|
>   |AD-PT|3%|52.01/43.99|58.51/57.85|50.22/32.52|47.31/41.59|
>   |Scratch|20%|57.85/50.69|62.18/61.64|58.18/40.64|53.18/49.80|
>   |AD-PT|20%|59.71/53.49|64.10/63.54|59.00/43.13|56.04/53.80|
>
>   |Methods (PointRCNN)|Data amount (KITTI)|Overall|Car|Ped.|Cyc.|
>   |-|-|-|-|-|-|
>   |Scratch|20%|64.12|75.30|52.52|69.55|
>   |AD-PT|20%|67.67|77.20|54.16|71.86|
>   |Scratch|100%|68.40|78.70|54.41|72.11|
>   |AD-PT|100%|70.47|80.25|57.13|74.04|
>
>   We conduct point-based experiments on KITTI dataset for a fair comparison due to the lack of config for Waymo in OpenPCDet.
>   The fine-tuning performance of point- and pillar-based methods can be further improved, when the models are initialized by our pre-trained checkpoint, showing that our method can combine with multiple types of 3D detectors.
> - We agree with the reviewer's suggestion that we do not provide a general backbone but provide a general pre-training pipeline that can be used on various types of 3D detectors. **We will revise it in the next version**.
>
> **Q2: About the voxel size used in pre-training.**
>
> **A2**:
> - The voxel size used in the experiments.
>   |Dataset (setting)|Voxel size|
>   |-|-|
>   |ONCE (pre-training)|[0.1,0.1,0.2]|
>   |Waymo (Fine-tuning)|[0.1,0.1,0.15]|
>   |nuScenes (Fine-tuning)|[0.1,0.1,0.2]|
>   |KITTI (Fine-tuning)|[0.05,0.05,0.1]|
>   As shown in the table, the voxel size used in pre-training is different from that used in the downstream task (fine-tuning stage). Especially for KITTI dataset, the table shows the flexibility of loading the backbone with different voxel sizes.
> - For fair comparison to previous methods like BEV-MAE, we use [0.1,0.1,0.2] on nuScenes in the main text. Further, we also supplement the results, by conducting the experiment on nuScenes using [0.075,0.075,0.2] in the following table.
>   |Methods (CenterPoint)|mAP|NDS|
>   |-|-|-|
>   |Scratch|58.07|66.00|
>   |AD-PT|59.30|66.72|
>
> **Q3: I want to know whether extending the training schedule can achieve comparable performance or not.**
>
> **A3**:
> - First, we conduct experiments on extending the training schedule, and the results are shown as follows.
> |Methods (PV-RCNN++)|Data amount (Waymo)|Epochs|Overall L2 AP/APH|Veh.|Ped.|Cyc.|time cost|
> |-|-|-|-|-|-|-|-|
> |Scratch|20%|150|70.95/68.51|70.25/69.81|72.18/66.35|70.43/69.38|5x time|
> |AD-PT|20%|30|71.55/69.2|70.62/70.19|72.36/66.82|71.69/70.70|1x time|
> |Scratch|3%|150|68.76/66.18|68.10/67.62|70.08/63.88|68.11/67.03|5x time|
> |AD-PT|3%|150|69.72/67.14|69.00/68.52 |71.11/64.93|69.04/67.96|5x time|
>
> Initialized by our pre-trained model, the results of only training 30 epochs can exceed the results of 150 epochs of training from scratch. Besides, the performance using our pre-trained model can consistently outperform train from scratch when extending the training schedule (e.g., 3% fine-tuning data for 150 epochs).
> - Further, our method can easily scale up the pre-training data compared with previous methods, and the performance of the downstream dataset can be further improved as shown in table 8 and the following table.
>   |Methods (SECOND)|Overall L2 AP/APH| Veh.|Ped.|Cyc.|
>   |-|-|-|-|-|
>   |Scratch|60.62/56.86|64.26/63.73|59.72/50.38|57.87/56.48|
>   |AD-PT (100K pre-training data)|61.26/57.69|64.54/64.00|60.25/51.21|59. 00/57.86|
>   |AD-PT (500K pre-training data)|62.34/58.74|65.20/64.66|61.26/52.20|60. 56/59.36|
>   |improvement|+1.72/+1.88|+0.94/+0.93|+1.54/+1.82|+2.79/+2.88|
>
> **Q4: Difference between our proposed AD-PT and pseudo-labels based 3D semi-supervised learning.**
>
> **A4**：
> - *Task-level differences*: Different from pseudo-labeling and semi-supervised learning mentioned by the reviewer which aims to find a better pseudo-labeling method, our work aims to propose a *pre-training* method that can learn unified 3D representations that can improve the performance on multiple downstream datasets. Meanwhile, previous 3D pre-training methods mainly focus on pre-training and fine-tuning on the same dataset. AD-PT is the first work to focus on pre-training and evaluating on different downstream datasets (Waymo, KITTI, nuScenes) in this community.
> - *Algorithm-level differences*:
>   - Pseudo-labeling mentioned by the reviewer tries to generate more useful pseudo labels using semi-supervised learning methods. As mentioned in our Method Section, we use the semi-supervised learning method to improve the quality of pseudo labels, which is **only a part** of our pre-training pipeline.
>   - Our proposed method mainly considers the generalization of the model while improving the performance of downstream tasks which is totally different from previous pre-training methods and semi-supervised learning methods.
>     - The re-scaling and re-sampling methods aim to increase the diversity of pre-trained data which improves the backbone's generalization ability at data-level.
>     - Unknown-aware instance learning head considers the difference between pre-training and fine-tuning dataset which improves the backbone's generalization ability at algorithm-level.
> - *Application-level differences*: As the amount of pre-training data increases, the performance of downstream tasks will be continuously improved as shown in Table 8 and table in **A3**. Such a phenomenon **cannot be observed** by previous 3D AD pre-training methods.

---

> > ### Comment · Reviewer_nLKe · 2023-08-16
> > **Final Rating**
> >
> > After reading the comments from other reviews and the rebuttal from the authors, I still think the novelty of this work is limited. However, I appreciate the additional experiments which shows the generalization ability of this work. So I'd like to slightly improve my score to 'borderline accept'

---

> > > ### Author Response · Authors · 2023-08-17
> > > **Thanks for your approval, and clarify our novelty**
> > >
> > > Thank you very much for your approval of AD-PT's generalization ability. We appreciate the reviewer's comments on our works. Due to the limited number of characters in the rebuttal phase, we would like to clarify the novelty of AD-PT from the following three aspects.
> > >
> > > **Motivation-level**: AD-PT is the first work aiming at using totally different pre-training and fine-tuning datasets and the first work that can provide backbones with strong generalization ability. Such a new pre-training paradigm can bring the following three advantages.
> > >
> > >  - Data-efficient. Previous works mainly follow paradigms that use **the same dataset** for pre-training and fine-tuning. However, when the sensor is updated, the previous pre-training model can not improve the performance of the new data. In contrast, AD-PT can provide backbones with strong generalization ability, which means that the previous pre-trained backbone can improve the performance of new data. Meanwhile, in the rebuttal phase, we show that our method can easily combine with multi-dataset pre-training, which means we can use data from old sensors. **In conclusion, AD-PT can reuse previous data to provide a good pre-training model without re-collecting a large amount of data from the new sensor**.
> > >
> > > - Training-efficient. Previous methods need to pre-train new backbone parameters using new data when we want to improve the performance of newly collected data, which is time-consuming. **In contrast, AD-PT can achieve a one-shot pre-training to improve the performance of multiple domains, greatly reducing pre-training time and computation power consumption**.
> > >
> > > - Scaling up performance. Limited by the amount of data, previous works are hard to observe the performance gains when increasing the pre-training data. However, **AD-PT verifies that the downstream performance is positively correlated with the amount of the upstream pre-training data**.
> > >
> > > **Framework-level**: AD-PT is the first work to use semi-supervised pre-training and provides a complete set of pre-training processes including data labeling and training algorithm process  (We will release the large-scale pseudo-labeled point clouds, about 1M data, which is the largest-scale labeled outdoor point-cloud dataset). Meanwhile, the starting point of AD-PT is totally different from previous semi-supervised learning, and data augmentation works as follows.
> > >
> > > - Previous pre-training methods mainly use MAE-based or contrastive learning-based methods to obtain backbone parameters to achieve the best in the single domain, while AD-PT explores a new pseudo-label-based method and totally considers the differences between upstream and downstream datasets, such as the differences in semantic classes between upstream and downstream. As a result, we propose to Unknown-aware Instance Learning Head to perform the open-set detection on the downstream dataset.
> > >
> > > - Furthermore, we provide a complete pre-training process including data preparation and a pre-training algorithm that is totally different from the previous pre-training methods as shown in Fig. 2.
> > >
> > > **Component-level**: Finally, each proposed module is also different from the existing ones.
> > >
> > > - Data preparation. Based on the observation that the quality of pseudo labels can affect the performance of pre-training process. We propose to use the class-aware pseudo generator and semi-supervised learning to obtain as accurate pseudo labels as possible. Pseudo-labeling as mentioned by the reviewer improves semi-supervised performance by using pseudo-labeling by means of threshold which is only a part of our method. Besides, our re-scaling and re-sampling methods meanly consider improving the diversity of pre-training data, which is different from most previous data augmentation methods aiming at improving the performance of the single-domain model.
> > >
> > > - Training algorithm. AD-PT is the first work that introduces the idea of open-set learning to the pre-training task, which fully considers the taxonomy difference between upstream and downstream datasets. We found some negative samples, that could be ignored during the upstream pre-training period, may be useful for different downstream fine-tuning tasks (due to the taxonomy differences between datasets).  As a result, we consider such samples as undetected instances. Such an idea is different from previous MAE-based and contrastive-learning-based methods.

---

### Official Review · Reviewer_hHpC · 2023-07-02

**Soundness:** 2 fair
**Presentation:** 3 good
**Contribution:** 2 fair
**Rating:** 5
**Confidence:** 4

**Summary:**

The paper proposes an autonomous driving pre-training method, where a large-scale pre-training dataset is built and a generalizable representation is created. To further improve the performances on the down-stream tasks, an advanced pre-training technique of semi-supervised method is proposed. Moreover, an unknown-aware instance learning head is proposed to learn the open-set detection. Various experiments are conducted to show the effectiveness of the method.

**Strengths:**

1. The paper is well-organized and clearly written.
2. Thorough experiments are conducted on various benchmarks.

**Weaknesses:**

1. The novelty is a bit limited, since the proposed techniques look like a combination of existing approaches, such as data augmentation and semi-supervised learning.
2. Although the improvements on various benchmarks are promising, the results are inferior compared to SOTA Lidar based object detection approaches such as VISTA [A] and MGTANet [B] on the nuScenes dataset. Hence, it would be great to also test on current SOTA approaches, which could further verify the generalization ability and robustness of the method.
3. Eq. 2 seems to be incorrect since x and y share the same calculation. It would be further clearer if a figure is drawn to show the calculation.
4. Line 293-294: Should the improvements for Waymo and nuScenes be 0.65%/0.54% and 7.37%, respectively?

[A]: Deng, Shengheng, et al. "Vista: Boosting 3d object detection via dual cross-view spatial attention." Proceedings of the IEEE/CVF Conference on Computer Vision and Pattern Recognition. 2022.
[B]: Koh, Junho, et al. "MGTANet: Encoding Sequential LiDAR Points Using Long Short-Term Motion-Guided Temporal Attention for 3D Object Detection." Proceedings of the AAAI Conference on Artificial Intelligence. Vol. 37. No. 1. 2023.

**Questions:**

None

**Limitations:**

The limitation discussion seems to be missing. Possible memory consumption or time efficiency could be discussed to show a more comprehensive comparison with other SOTA works.,

---

> ### Author Rebuttal · Authors · 2023-08-09
>
> ## Response to Reviewer hHpC
>
> We sincerely appreciate the Reviewer's efforts and comments. We have also tried our best to clarify the novelty of the proposed method and supplement more experimental results and discussions here.
>
> **Q1: Novelty is a bit limited, since the proposed techniques look like a combination of existing approaches, such as data augmentation and semi-supervised learning.**
>
> **A1**:
>
> - Our study provides the autonomous driving community with **knowledge** that scaling up the amount of pre-training samples can boost the performance on **multiple downstream datasets** simultaneously. Such an insight can inspire the pre-training study in the autonomous driving community, with the purpose of fully leveraging the constantly increasing autonomous driving data.
> - Our proposed method mainly considers the generalization ability of the model while improving the performance of downstream tasks.
>   - The re-scaling and re-sampling methods aim to increase the diversity of pre-trained data which improves the backbone's generalization ability in **data-level**.
>   - Unknown-aware instance learning head fully considers the difference of pre-training and fine-tuning dataset which improves the backbone's generalization ability in **algorithm-level**.
>
>   As shown in Tables 8, 9, 10 of the main text, initialized by our pre-trained backbone (**a single checkpoint**), the performance of multiple downstream datasets can be improved and even surpass the baselines which pre-train and fine-tune on the same dataset.
>
> - Further, *as the amount of pre-trained data increases, the performance of downstream tasks will be continuously improved as shown in Table 8* of the main text. Such a phenomenon **cannot be observed** by previous pre-training methods (*e.g.* ProposalContrast, BEV-MAE, etc.), due to that the scale of the pre-training dataset is difficult to be continuously expanded, only using previous pre-training methods (they perform the pre-training and fine-tuning on the same benchmark). Besides, we conduct experiments on more fine-tuning data (*i.e.* Waymo 20%), as shown in the following table. Overall, different from previous semi-supervised learning and 3D pre-training methods (*e.g.* ProposalContrast, BEV-MAE, etc.), which aim to improve the performance **on the same dataset**, our work aims to perform the pre-training process and downstream fine-tuning process **across different datasets**, reducing the model retraining cost and achieving the fine-tuning performance scalability. This is the first work to focus on pre-training 3D backbone that can be verified to be effective in many autonomous driving datasets such Waymo, nuScenes, KITTI.
>
>   |Method| Overall L2 AP / APH | Vehicle| Pedestrian | Cyclist|
>   | --- | --- | --- | --- | --- |
>   |AD-PT (SECOND, 100K pre-trained data) | 61.26 / 57.69 | 64.54 / 64.00 | 60.25 / 51.21 | 59.  00 / 57.86 |
>   |AD-PT (SECOND, **500K** pre-trained data) | 62.34 / 58.74 | 65.20 / 64.66| 61.26 / 52.20 | 60. 56 / 59.36 |
>
> - Current semi-supervised learning methods such as MeanTeacher cannot achieve a significant performance gain towards the downstream dataset, as shown in Table 1 of the main text and the following table.
>
>   |Method| mAP| NDS|
>   | --- | --- |---|
>   |MeanTeacher | 55.46 | 63.93 |
>   |AD-PT | 57.17 | 65.48 |
>
> **Q2: Although the improvements on various benchmarks are promising, the results are inferior compared to SOTA Lidar based object detection approaches such as VISTA.**
>
> **A2**: Thanks for your valuable comment. According to this Reviewer's suggestion, we supplement the experiments of initializing VISTA [Ref-A] with our pre-trained model using AD-PT. The result of training VISTA from scratch is 60.8 / 68.1 (as reported in their original paper [Ref-A]), and the result of training VISTA using the AD-PT checkpoint is 61.12 / 68.38. We report the results on nuScenes validation set, and the experimental results demonstrate that VISTA can be further improved by pre-training using the AD-PT method, which further verifies the generalization ability of our method. Furthermore, we will add the comparison results with VISTA [Ref-A] **in the next version**.
>
> [Ref-A]  Deng, Shengheng, et al. "Vista: Boosting 3d object detection via dual cross-view spatial attention." Proceedings of the IEEE/CVF Conference on Computer Vision and Pattern Recognition. 2022.
>
> **Q3: Eq. 2 seems to be incorrect since x and y share the same calculation. It would be further clearer if a figure is drawn to show the calculation.**
>
> **A3**: We thank the Reviewer very much for pointing it out. The correct formula should be:
>
> $
> x=rcos(\phi)cos(\theta),  y=rcos(\phi)sin(\theta), z=rsin(\phi)
> $
>
> We will correct the formula in the next version and provide a figure to clearly show the calculation in the **one-page PDF**.
>
>
> **Q4: Line 293-294: Should the improvements for Waymo and nuScenes be 0.65%/0.54% and 7.37%, respectively?**
>
> **A4**: Thank the Reviewer very much for this comment, and we will correct it in the next version.
>
> Limitation: Although the AD-PT pre-trained backbone can improve the performance on multiple downstream datasets, it needs to be verified in more actual road scenarios. Meanwhile, training a backbone with more generalization capabilities through data from different sensors is also a future direction.
>
> We will add a limitation section to our next version.

---

> > ### Comment · Reviewer_hHpC · 2023-08-15
> >
> > Thanks for the rebuttal.
> > To make the comparison to existing works fair enough, it is highly encouraged to directly compare to the widely used nuScenes testing data shown in Tab.1 of VISTA [A] and MGTANet [B] to show the effectiveness of the proposed method.

---

> > > ### Author Response · Authors · 2023-08-17
> > > **Thanks for your suggestion, and the nuScenes test results**
> > >
> > > We would like to thank the reviewer for this valuable suggestion. According to the reviewer's comment, we further submit the nuScenes test results of VISTA initialized by our proposed AD-PT to the **nuScenes server** to make a fair comparison. We follow VISTA to use double flip for testing augmentation. The test performance is shown below, where we use the performance of VISTA-OHS reported in their original paper for a fair comparison.
> > >
> > > | Methods | NDS | mAP | car | truck | cons. | bus | trailer | barrier | motorcycle | bicycle | pedestrian | traffic cone|
> > > | --- | --- | --- | --- | --- | --- | --- | --- | --- | --- | --- | --- | --- |
> > > |VISTA-OHS (from scratch) | 69.8 | 63.0 | 84.4 | 55.1 | 25.1 | 63.7 | 54.2 | 71.4 | 70.0 | 45.4 | 82.8 | 78.5 |
> > > |VISTA-OHS (AD-PT) | 70.47 | 63.84 | 84.6 |  54.1 | 29.0 | 64.3 | 55.3 | 71.3 | 71.2 | 45.4 | 83.7 | 78.9 |
> > >
> > > We are very grateful for this insightful comment and will add the comparison of VISTA-OHS mentioned by the reviewer in the next version.
> > >
> > > Furthermore, we would like to emphasize that, the checkpoint obtained by our proposed AD-PT boosts the model performance not only on the nuScenes dataset (includes validation set and test set) but also on other public datasets, such as Waymo dataset (1.65% gain on PV-RCNN++ as shown in Table 3 of the main text), KITTI dataset (2.44% gain on PV-RCNN as shown in Table 5 of the main text).

---

> > > > ### Comment · Reviewer_hHpC · 2023-08-19
> > > >
> > > > Thanks for the reply and my concerns are addressed. I would like to change my rating to 'borderline accept'.

---

### Official Review · Reviewer_8so5 · 2023-07-06

**Soundness:** 3 good
**Presentation:** 3 good
**Contribution:** 3 good
**Rating:** 5
**Confidence:** 4

**Summary:**

This paper aims at large-scale point cloud dataset pre-training. The authors propose AD-PT method to build pre-training dataset  with diverse data distribution and learn generalizable representations. In details, they design a diversity-based pre-training data preparation procedure and unknown-aware instance learning. The extensive experimental results on Waymo, nuScenes and KITTI datasets verify the effectiveness of the proposed method.

**Strengths:**

1. Pre-training on the large-scale point cloud dataset is meaningful for autonomous driving. On the one hand, human annotation is expensive and how to use unlabeled data is an improtant direction. On the other hand, the point clouds of different LiDAR sensors have different patterns and we need consider these domain gaps.
2. The proposed method is reasonable. Although some thechniques are not new, suach as contrastive learning, I think they are reasonable for large-scale point cloud pretraining. For example, the LiDAR beam re-sampling can increase the robustness of beam domain gap.
3. The paper is well-written and easy to read.

**Weaknesses:**


1. The results in Table 3 cannot show the superiority of the proposed method. The performance gain with SS-PT methods such as BEV-MAE is marginal.

 2. I think a more useful setting is that we have some datasets A,B and C collected from different LiDAR sensors. We can use the labels in all three datasets and we want to improve the detector's performance on C dataset. Can the proposed method be applied to this setting? I hope the authors can give some insights.

 3. The authors should give an analysis between the detector's performance on finetuning dataset and the labels amount in pre-training datasets. As an extreme case, if we do not have labels in pre-training dataset, will the performance drop largely?

**Questions:**

Please reply to the weakness part.

**Limitations:**

The paper does not have limitation section.

---

> ### Author Rebuttal · Authors · 2023-08-09
>
> ## Response to Reviewer 8so5
>
> Dear Reviewer XpxU,
>
> Thanks for your review, we provide more experimental results and explanations of your question.
>
> **Q1: Table 3 cannot show the superiority of the proposed method. The performance gain with SS-PT methods such as BEV-MAE is marginal.**
>
> **A1**:
> - Different from previous pre-training methods like BEV-MAE, which use **the same** dataset as the pre-training and fine-tuning dataset. AD-PT mainly focuses on designing a general pre-training pipeline that can pre-train backbones with strong generalization ability which can improve the performance on **multiple downstream datasets** (e.g., nuScenes, Waymo).
>
> - Here we list the percentage of our performance improvement over BEV-MAE compared to training from scratch ($(ADPT - BEVMAE) / (BEVMAE - Scratch)$). As shown in the following table, the percentage of Waymo's improvement is more than 45% compared with BEV-MAE under multiple detectors.
>
>   |Detector|Percentage (AP / APH) |
>   | --- | --- |
>   |SECOND|  56.1% / 88.6%   |
>   |CenterPoint|  55.6% / 45.4%  |
>   |PV-RCNN++| 198.1% /  211.3%  |
>
>  - Meanwhile, using the same AD-PT pre-trained checkpoint, the performance on multiple datasets can be improved (e.g., nuScenes). However, under cross-dataset settings (i.e., Waymo pre-train, nuScenes fine-tune), performance can not be improved when initialized by BEV-MAE, as shown in the following table.
>   |Method|mAP|NDS|
>   | ---  | ---| --- |
>   |From scratch (CenterPoint) | 56.2 | 64.5|
>   |BEV-MAE (CenterPoint) | 56.30| 64.62|
>   |AD-PT (CenterPoint) | 57.17 | 65.48 |
>
> - Further, AD-PT achieves continuous performance gains, by expanding the scale of the pre-training dataset. When pre-training on a more large-scale dataset (i.e., 500K ONCE samples), the performance can be further improved as shown in the following table.
>
>   |Method| Overall L2 AP / APH | Vehicle| Pedestrian | Cyclist|
>   | --- | --- | --- | --- | --- |
>   |BEV-MAE (SECOND) | 61.03 / 57.30 | 64.42 / 63.87| 59.97 / 50.65| 58.69 / 57.39 |
>   |AD-PT (SECOND, 100K pre-trained data) | 61.26 / 57.69| 64.54 / 64.00 | 60.25 / 51.21 | 59.  00 / 57.86 |
>   |AD-PT (SECOND, 500K pre-trained data) | 62.34 / 58.74| 65.20 / 64.66| 61.26 / 52.20 | 60. 56 / 59.36 |
>
> - When fine-tuning on a small amount of data, AD-PT shows better performance compared to previous methods.
>
>   |Method | Data amount | Overall L2 AP / APH | Vehicle| Pedestrian | Cyclist|
>   | ---|---|---|---|---|---|
>   |From scratch (PV-RCNN++) | 3% | 63.81 / 61.10 | 64.42 / 63.93 | 64.33 / 57.79 | 62.69 / 61.59 |
>   |BEV-MAE (PV-RCNN++) | 3% | 64.87 / 62.05 | 65.54 / 65.04 | 65.46 / 58.98 | 63.62 / 62.15|
>   |AD-PT (PV-RCNN++) | 3% | 68.33 / 65.69 | 68.17 / 67.70 | 68.82 / 62.39 | 68.00 / 67.00|
>
> **Q2: ...We have some datasets A,B and C collected from different LiDAR sensors. ... Can the proposed method be applied to this setting?**
>
> **A2**: Thanks for your constructive suggestion, data from different sensors can greatly improve the diversity of the pre-training dataset. AD-PT proposes a method to increase the diversity of data with almost zero cost and an effective pre-training method. When obtaining more datasets with different sensors, the AD-PT can also effectively increase the diversity and our pre-trained method can easily be combined with different datasets.
>
> We conduct simple experiments to show that our method can be used in **multi-dataset pretraining scenarios**. We use KITTI dataset, ONCE labeled dataset and our pseudo-labeled dataset. KITTI dataset is collected by a 64-beam LiDAR and ONCE dataset is collected by a 40-beam LiDAR. Inspired by Uni3D[1], we align the point cloud range. The following table shows the fine-tuning results on 3% Waymo train set.
>
>   |Method|Fune-tuning data amount|Pre-training data |Overall L2 AP / APH |Vehicle|Pedestrian |Cyclist|
>   | ---|---|---|---|---|---| ---|
>   |AD-PT (PV-RCNN++) | 3% | once labeled dataset, pseudo-labeled dataset |68.33/65.69|68.17/67.70|68.82/62.39|68.00/67.00|
>   |AD-PT (PV-RCNN++) | 3% | KITTI dataset, once labeled dataset, pseudo-labeled dataset |68.67/66.00|68.47/67.98|69.12/62.63|68.41/67.39|
>
>
> [1] Zhang B, Yuan J, Shi B, et al. Uni3d: A unified baseline for multi-dataset 3d object detection[C]//Proceedings of the IEEE/CVF Conference on Computer Vision and Pattern Recognition. 2023: 9253-9262.
>
> **Q3:  Give an analysis between the detector's performance on finetuning dataset and label amount in pre-training datasets.**
>
> **A3**:
> - AD-PT uses the semi-supervised manner to pseudo a large amount of unlabeled data. Here, we conduct experiments on reducing the labeled data, and the results are shown in the following table. The ONCE labeled set consists of a total of 6 sequences, and we reduce the amount of data for labeling by extracting random sequences.
>
>   |Method| Labeled data | Overall L2 AP / APH | Vehicle| Pedestrian | Cyclist|
>   | --- | --- | --- | --- | --- | --- |
>   |AD-PT|  1 sequence from labeled set | 67.11/64.72 | 66.98/66.48 | 67.93/61.29 | 66.44/65.41 |
>   |AD-PT| 3 sequences from labeled set | 67.47 / 64.78 | 67.52/67.03 |68.35/61.80|66.54/65.52|
>   |AD-PT| 6 sequences from labeled set | 68.33 / 65.69 | 68.17/67.70 | 68.82/62.39 | 68.00/67.00 |
>
> It can be seen that performance degrades as the amount of annotated data decreases. Such a degradation is mainly due to that the accuracy of pseudo-labels will drop and such phenomenon is consistent with Figure 6 of the main text. Besides, Table 8 in the main text also shows the effect when using different amounts of unlabeled data.
>
> **Limitation**: Although the AD-PT pre-trained backbone can improve the performance on multiple downstream datasets, it needs to be verified in more actual road scenarios. Meanwhile, training a backbone with more generalization capabilities through data from different sensors is also a future direction.
> We will add a limitation section to our next version.

---

> > ### Comment · Reviewer_8so5 · 2023-08-20
> > **Keep my positive rating**
> >
> > Thanks to the author's thoughtful response, I feel that my questions have been mostly resolved, and I will maintain my rather positive rating.

---

### Author Rebuttal · Authors · 2023-08-09

## General Response:

Dear AC and reviewers,

Many thanks for your valuable comments and constructive suggestions to improve the quality of our work. Here is a summary of what we have done in the rebuttal phase.

- We conduct several new experiments to cover the concerns of the reviewers.

  - Verifying the effectiveness of our pre-training methods on different types of 3D backbones (e.g., Pillar-based backbone, point-based backbone)

  - Compare our results with the result of simply extending the training schedule

  - Applying our pre-trained backbone to more SOTA works (i.e., VISTA)

  - More experiments to show the sensitivity of voxel size

  - A simple attempt at a fusion of multiple datasets

   - Reducing the labeled data

  - Fine-tuning on 100% Waymo data

  - More experiments to show the generalization ability of pre-trained backbone obtained by AD-PT

  - More experiments to show the fine-tuning performance when scaling pre-training data

- We further provide more discussion on our insight and novelty, where we provide the autonomous driving community with the knowledge that scaling up the amount of pre-training samples can boost the performance on multiple downstream datasets simultaneously. Such an insight can inspire the pre-training study in the autonomous driving community, with the purpose of fully leveraging the constantly increasing autonomous driving data.

- We provide a discussion on the performance of our methods and previous methods.

- We provide a discussion on how to train on the dataset with multiple sensor types.

- We provide the limitation of our methods.

Thank you again for your precious time on the review. We hope that our response has addressed your concerns. We are happy to have further discussion on anything unclear about our paper.

Best regards,
Authors of paper 4632

---

### Decision · Program_Chairs · 2023-09-21

**Decision:**

Accept (poster)

**Comment:**

All reviewers agree that the paper tackles an important problem and proposes a solution with demonstrated effectiveness. There are some concerns that individual components of the approach is not novel, but reviewers agree that the paper should be accepted nevertheless because of strong results.